# Machine Learning for numerical weather and climate modelling: a review

Catherine O. de Burgh-Day[1] & Tennessee Leeuwenburg[1]

[1]The Bureau of Meteorology, 700 Collins St Docklands, Victoria, Australia

*Correspondence to:* Catherine O. de Burgh-Day (catherine.deburgh-day@bom.gov.au)

**Abstract.**

Machine learning (ML) is increasing in popularity in the field of weather and climate modelling. Applications range from improved solvers and preconditioners, to parameterization scheme emulation and replacement, and more recently even to full ML-based weather and climate prediction models. While ML has been used in this space for more than 25 years, it is only in the last 10 or so years that progress has accelerated to the point that ML applications are becoming competitive with numerical knowledge-based alternatives. In this review, we provide a roughly chronological summary of the application of ML to aspects of weather and climate modelling from early publications through to the latest progress at the time of writing. We also provide an overview of key ML terms, methodologies, and ethical considerations. Finally, we discuss some potentially beneficial future research directions. Our aim is to provide a primer for researchers and model developers to rapidly familiarize and update themselves with the world of ML in the context of weather and climate models.

## 1. Introduction

Current state-of-the-art weather and climate models use numerical methods to solve equations representing the dynamics of the atmosphere and ocean on meshed grids. The grid-scale effects of processes that are too small to be resolved are either represented by parametrization schemes or are prescribed. These numerical weather and climate forecasts are computationally costly and are not easy to implement on specialized compute resources such as GPUs (although there are efforts underway to do so, for example in LFRic (Adams et al. 2019)). One of the main approaches to improving forecast accuracy is to increase model resolution (reduced timestep between model increments and/or decreased grid spacing), but due to the high computational cost of this approach, improvements in model skill are hampered by the finite supercomputer capacity available. An additional pathway to improve skill is to improve the understanding and representation of subgrid-scale processes, however this is again a potentially computationally costly exercise.

In the remainder of this introduction, we overview the state of machine learning in weather and climate research without always providing references; we instead provide relevant references in the detailed sections that follow.

Machine learning is an increasingly powerful and popular tool. It has proven to be computationally efficient, as well as being an accurate way to model subgrid-scale processes. The term "Machine learning" (ML) was first coined by

Arthur Samuel in 1952 to refer to a "field of study that gives computers the ability to learn without being explicitly
programmed"[1]. Learning by example is the defining characteristic of ML.
The growing potential for ML in weather and climate modelling is being increasingly recognized by meteorological
agencies and researchers around the world. The former is evidenced by the development of strategies and frameworks
to better support the development of ML research, such as the Data Science Framework recently published by the Met
Office in the UK[2]. The latter is made clear by the explosion in publications from academia, government agencies and
private industry in this space, as demonstrated by the rest of this review. Figure 1 shows the number of publications
cited in this review using different categories of ML algorithms by year, and clearly illustrates the increase in the
uptake of ML methods by the research community.
This is not necessarily an unbiased sample of the use of different architectures in the literature, since the selection of
papers cited in this review focuses on telling the story of the growth of the use of ML in weather and climate modelling
over time, rather than being a comprehensive list of all uses of ML in the literature.
There are established techniques and aspects of the weather and climate modelling lifecycle that would already be
considered ML by many. For example, linear regression[†3], principal component analysis, correlations, and the
calculation of teleconnections can all be considered types of ML. Data Assimilation techniques could also be
considered a form of ML. There are, however, other classes of ML (e.g. Neural Networks[†], Decision Trees[†], etc.)
which are much less widely used within the weather and climate modelling space and have great potential to be of
benefit. There is growing interest in, and increasingly effective application of, these ML techniques to take the place
of more traditional approaches to modelling. The potential for ML in weather and climate modelling extends all the
way from replacement of individual sub-components of the model (to improve accuracy and reduce computational
cost) to full replacement of the entire numerical model.
While ML models are typically computationally costly during training, they can provide very fast predictions at
inference[†] time, especially on GPU hardware. They often also avoid the need to have full understanding of the
processes being represented and can learn and infer complex relationships without any need for them to be explicitly
encoded. These properties make ML an attractive alternative to traditional parametrization, numerical solver, and
modelling methods.
Neural Networks (NNs, explained further in Section 2.1) in particular are an increasingly favored alternative approach
for representing sub-grid-scale processes or replacing numerical models entirely. They consist of several
interconnected layers of nonlinear nodes[†], with the number of intermediate layers depending on the complexity of the
system being represented. These nodes allow for the encoding of an arbitrary number of interrelationships between
arbitrary parameters to represent the system, removing the need to explicitly encode these interrelationships into a
parameterization or numerical model.

---

[1] http://infolab.stanford.edu/pub/voy/museum/samuel.html, accessed 7th February 2023

[2] https://www.metoffice.gov.uk/research/foundation/informatics-lab/met-office-data-science-framework, accessed 7 February 2023

[3] Henceforth, the first occurrence of each term described in the glossary is marked with the symbol "†"

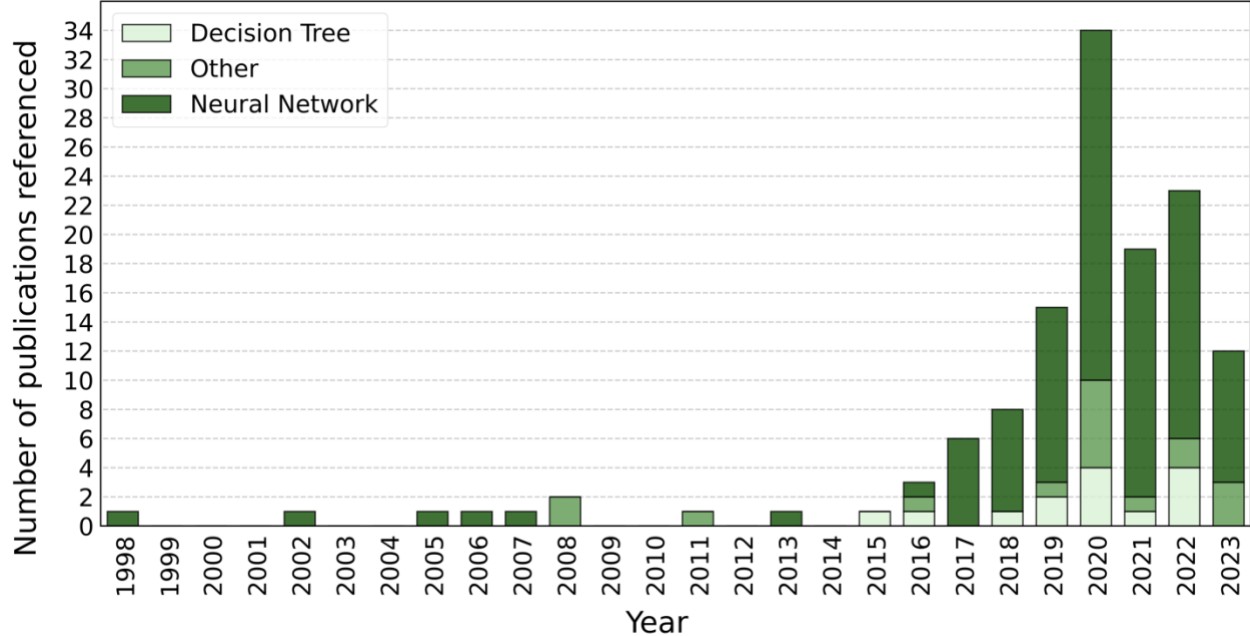

Figure 1: A stacked bar graph of the number of publications cited in this review using different categories of ML algorithms by per year. For a description of Neural Networks and Decision Trees see Section 2.1 and 2.2 respectively. The 'Other' category is a collection of ML model types other than decision trees and neural networks, each of which only had one or two examples of use in this review. This included custom supervised and self-supervised algorithms, support vector machines and relevance vector machine models, regression models, unsupervised learning models, reservoir computing models and non-NN gaussian models. This figure includes all references from this review except for: seminal ML papers that are on new ML methods (e.g., foundational ML papers from outside the domain of weather and climate modelling), review papers, any paper cited that concerns a topic which is out of scope (e.g., nowcasting), and any other paper which does not present a new method directly applicable to weather and climate modelling. The full table of citations is provided in the appendix.

One challenge that must be overcome before there will be more widespread acceptance of ML as an alternative to traditional modelling methods is that ML is seen as lacking interpretability. Most ML models do not explicitly represent the physical processes they are simulating, although physics constrained ML is a new and growing field which goes some way to addressing this (see Section 6). Furthermore, the techniques available to gain insight into the relative importance and predictive mechanism of each predictor (i.e. the model outputs) are limited. In contrast, traditional models are usually driven by some understanding and/or representation of the physical mechanisms and processes which are occurring. This makes it possible to more easily gain insight into what physical drivers could explain a given output. The "black box" nature of many current ML approaches to parametrization makes them an unpopular choice for many researchers (and can be off-putting for decision makers) since, for example, explaining what went wrong in a model after a bad forecast can be more challenging if there are processes in the model which are not, and cannot, be understood through the lens of physics. However, increasing attention is being paid to the interpretability of ML models (e.g., McGovern et al., 2019; Toms et al., 2020; Samek et al., 2021), and there are

existing methods to provide greater insight into the way physical information is propagated through them (e.g., attention maps, which identify the regions in spatial input data that have the greatest impact on the output field, and ablation studies, which involve comparing reduced data sources and/or models to the original models that have full access to available data, to gain insight into the models).

As with their traditional counterparts, ML-based parametrizations and emulators are typically initially developed in single-column models, aquaplanet configurations, or otherwise simplified models. There are many examples of ML-based schemes which have been shown to perform well against benchmark alternatives in this setting, only to fail to do so in a realistic model setting. A common theme is that these ML schemes rapidly excite instabilities in the model as errors in the ML parametrization push key parameters outside of the domain of the training data as the overall model is integrated forward in time, leading to rapidly escalating errors and to the model 'blowing up'. Similarly, many ML-based full model replacements perform well for short lead times, only to exhibit model drift and a rapid loss of skill for longer lead times due to rapidly growing errors and the model drifting outside its training envelope.

In recent years, however, progress has been made in developing ML parametrizations which are stable within realistic models (i.e. not toy models, aquaplanets etc.), and ML-based full models which can run stably and skillfully to longer lead times. This is usually achieved through training the model on more comprehensive data, employing ML architectures which keep the model outputs within physically real limits, or imposing physical constraints or conservation rules within the ML architecture or training loss functions[†].

There are still challenges and possible limitations to an ML approach to weather and climate modelling. In most cases, a robust ML model or parameterization scheme should be able to:

- remain stable in a full (i.e. non-idealized) model run,
- generalize to cases outside its training envelope,
- conserve energy and achieve the required closures.

Additionally, for an ML approach to be worthwhile it must provide one or more of the following benefits:

- For ML parametrization schemes:
  - a speedup of the representation of a subgrid-scale process vs. when run with a traditional parametrization scheme. This can make the difference between the scheme being cost-effective to run or not - when it is not cost-effective the process usually needs to be represented with a static forcing or boundary condition file,
  - a speedup of the model vs. when run with traditional parametrization schemes,
  - improved representation of sub-grid process(es) over traditional parameterization schemes, as measured by metrics appropriate to the situation,
  - improved overall accuracy/skill of the model when run with traditional parametrization schemes,
  - insight into physical processes not provided by current numerical models or theory.
- For full ML models:
  - a speedup of the model vs. an appropriate numerical model control,
  - improved overall accuracy/skill of the model vs. an appropriate numerical model control,
  - skillful prediction to greater lead times than an appropriate numerical model control,

o   insight into physical processes not provided by current numerical models or theory

Furthermore, in some cases of ML approaches to weather and climate modelling problems (particularly for full model
replacement) the work is led by data scientists and ML researchers with limited expertise in weather and climate model
evaluation. This can lead to flawed, misleading or incomplete evaluations. Hewamalage et al. (2022) have sought to
rectify this problem by providing a guide to forecast evaluation for data scientists.
The scope of this review is deliberately limited to the application of ML within numerical weather and climate models
or for their replacement. This is done to keep the length of this review manageable. ML has enormous utility for other
aspects of the forecast value chain such as observation quality assurance, data assimilation, model output
postprocessing, forecast/product generation, downscaling, impact prediction, decision support tools, etc. A review of
the application of, and progress in, ML in these areas would be of great value but is outside the scope of this review
and is left to other work. Molina et al. (2023) have provided a very useful review of ML for climate variability and
extremes which is highly complementary to this review. They draw similar lines of delineation in the earth system
modelling (ESM) value chain to those mentioned above; describing them as "initializing the ESM, running the ESM,
and postprocessing ESM output". They examine each of these steps in turn, with a focus on the prediction of climate
variability and extremes. Here we take a different approach, focusing on one part of the value chain (running the
ESM), but looking in more detail at this one part. Additionally, here we consider climate modelling in the context of
multiyear and free-running multidecadal simulations, but exclude the topic of ML for climate change projections,
climate scenarios, and multi-sector dynamics. This is again in the interests of ensuring the scope of the review is
manageable, rather than because these topics are not worthy of review. On the contrary, a review dedicated to the
utility of machine learning in this area would be of enormous value to the community, but cannot be adequately
explored here. A brief introduction to key ML architectures and concepts, including suggested foundational reading,
is also provided to aid readers who are unfamiliar with the subject.
The remainder of this review is structured as follows: In Section 2 an introduction to the two ML architectures most
prevalent in the review is provided, followed by a suggested methodological approach to applying ML to a problem,
and finally a brief overview of some of the major ML architectures and algorithms. With this background in place, the
application of ML in weather and climate modelling is explored in the following five sections: In Section 3, ML use
in sub-grid parametrization and emulation, along with tools and challenges specific to this domain, are covered.
Zooming out from subgrid-scale to processes resolved on the model grid, in Section 4 the application of ML for the
partial differential equations governing fluid flow is reviewed. Expanding scope further again to consider the entire
system, the use of ML for full model replacement or emulation is reviewed in Section 5. In Section 6 the growing field
of physics constrained ML models is introduced, and in Section 7 a number of topics tangential to the main focus of
this review are briefly mentioned. Setting the work covered in the previous sections in a broader context, a review of
the history of, and progress in, ML outside of the fields of weather and climate science is presented in Section 8. In
Section 9 some practical considerations for the integration of ML innovations into operational and climate models are
discussed, followed by a short introduction to some of the ethical considerations associated with the use of ML in
weather and climate modelling in Section 10. In Section 11, some future research directions are speculated on, and
some suggestions are made for promising areas for progression. Finally, a summary is presented in Section 12, and a
Glossary of Terms is provided after the final Section to aid the reader in their understanding of key concepts and
words.

## 2. A Quick Introduction to Machine Learning

While the scope of this paper is a review of ML work directly applicable to weather and climate modelling, an abridged
introduction to some key fundamental ML concepts is provided here to aid the reader. Suggested starting points for
interested readers, including guidance on the utility of different model architectures and algorithms, and the
connections between different applications and approaches, are as follows:

- • Hsieh (2023) provides a thorough textbook on environmental data science including statistics and machine
learning
- • Chase et al (2022a, 2022b) provide an introduction to various machine learning algorithms with worked
examples in a tutorial format and an excellent on-ramp to ML for weather and climate modelling
- • Russell & Norvig (2021) provide a comprehensive book regarding artificial intelligence in general
- • Goodfellow et al. (2016) provide a well-regarded book on deep learning theory and modern practise
- • Hastie et al. (2009) provide a book on statistics and machine learning theory

This introductory section is a brief exposition of the concepts most central to this review. Definitions for this section
can be found in the glossary.
The majority of ML methods which have found traction in weather and climate modelling were first developed in
fields such as computer vision, natural language processing and statistical modelling. Few, if any, of the methods
mentioned in this paper could be considered unique to weather and climate modelling, however, they have in many
cases been modified to a greater or lesser extent to suit the characteristics of the problem. In this review, the term
algorithm refers to the mathematical underpinnings of a machine learning approach. By this definition, decision trees
(DTs), NNs, linear regression and Fourier transforms are examples of algorithms. The two most relevant algorithms
for this review are DTs and NNs. Many ML algorithms can be thought of as optimizing a nonlinear regression, with
deep learning utilizing an extremely high-dimensional model. There is no consensus on the definition of ML, with the
term encompassing relatively large or small topical domains depending on who is asked. A good rule of thumb,
however, is that any iterative computational process that seeks to minimize a loss function or optimize an objective
function can be considered to be a form of ML. Some of the chief concerns in machine learning are generalizability
of the models, how to train (optimise the variables of) the model, and how to ensure robustness. The inputs and outputs
of machine learning models are the often same as physical models or model components.The term architecture in
machine learning refers to a specific way of utilizing an algorithm to achieve a modelling objective reliably. For
example, the U-Net[†] architecture is a specific way of laying out a NN which has proven effective in many applications.
The extreme gradient boosting decision tree[†] architecture is a specific way of utilizing DTs which has proven reliable
and effective for an extraordinary number of problems and situations and is an excellent choice as a first tool to
experiment with machine learning.
A major current focus of ML research in the context of weather and climate modelling is new NN-based architectures
and algorithms, and improved training regimes. Many other algorithms have been and continue to be employed in
machine learning more broadly, but are not pertinent to this review.
A key point for ML researchers to be aware of is the critical importance of approaching model training carefully.
There are many pitfalls which can result in underperformance, unexpected bias or misclassification. For instance,
adversarial examples[†] can occur 'naturally', and systems which process data can be subject to adversarial attack[†]
through the intentional supply of data designed to fool a trained network.
**2.1. Introduction to Neural Networks**
NNs can be regarded as universal function approximators (Hornik et al., 1989; see also Lu et al., 2019). Further, NN
architectures can theoretically be themselves modelled as a very wide feed-forward[†] NN with a single hidden layer.
A Fourier transform is another example of a function approximator, although it is not universal since not all functions
are periodic. NNs can therefore theoretically be candidates for accurate modelling of physical processes, although in
practise they cannot always reliably interpolate beyond their training envelope and as such may not generalize to new
regimes.ML models are typically introduced in the literature as being either classification[†] or regression[†] models, and
either supervised[†] or unsupervised[†].
The mathematical underpinning of a NN can be considered distinctly in terms of its evaluation[†] (i.e., output, or
prediction) step and its training update step. The prediction step can be considered as the evaluation of a many-
dimensional arbitrarily complex function.
The simplest NN is a single-input, single node network with a simple activation[†] function. A commonly used activation
function for a single neuron is the sigmoid function, which helpfully compresses the range between 0 and 1 while
allowing a nonlinear response. A classification model will employ a threshold to map the output into the target
categories. A regression model seeks to optimize the output result against some target value for the function. Larger
networks make more use of linear activations and may utilise heterogenous activation function choices at different
layers.
Complex NNs are built up from many individual nodes, which may have heterogenous activation functions and a
complex connectome[†]. The forward pass[†], by which inputs are fed into the network and evaluated against activation
functions to produce the final prediction, uses computationally efficient processes to quickly produce the result.
The training step for a NN is far more complex. The earliest NNs were designed by hand rather than through
automation. The training step applies a back-propagation[†] algorithm to apply adjustment factors to the weights[†] and
biases[†] of each node based on the accuracy of the overall prediction from the network.
Training very large networks was initially impractical. Both hardware and architecture advances have changed this,
resulting in the significant increase in application of NNs to practical problems. Most NN research explores how to
utilize different architectures to train more effective networks. There is little research going into improving the
prediction step as the effectiveness of a network is limited by its ability to learn rather than its ability to predict. Some
research into computational efficiency is relevant to the predictive step. NNs can still be technically challenging to
work with, and a lot of skill and knowledge are needed to approach new applications.
The major classes of NN architectures most likely to be encountered are:
• Small, fully-connected networks, which are less commonly featured in recent publications but are still
effective for many tasks and are still being applied and may well be encountered in practice
• Convolutional[†] architectures, first applied to image content recognition, which match the connectome of the
network to the fine structure of images in hierarchical fashion to learn to recognize high-level objects in
images
• Recurrent token-sequence architectures, first applied to natural language processing, generation and
translation; applicable to any time-series problem. Now also applied to image and video applications, and
mixed-mode applications such as text-to-image or text-to-video
• Transformer architectures[†], based on the attention mechanism[†] to provide a non-recurrent architecture which
can be trained using parallelized training strategies. This allows larger models to be trained. Originally
developed for sequence prediction and extended to image processed through vision transformer architectures.
**2.2. Introduction to Decision Trees**
DTs are a series of decision points, typically represented in binary fashion based on a simple threshold. A particular
DT of a particular size maps the input conditions into a final 'leaf' node which represents the outcome of the decisions
up to that point.
A random forest[†] (RF) is the composition of a large number of DTs assembled according to a prescribed generation
scheme, which are used as an ensemble. A gradient boosted decision tree (GBDT) is built up sequentially, where each
subsequent decision tree attempts to model the errors of the stack of trees built up thus far. This approach outperforms
RFs in most cases.
The DT family of ML architectures are very easy to train and are very efficient. They are well documented in the
public domain and in published literature. DTs are statistical in nature and are not capable of effectively generalizing
to situations which are not similar to those seen during training. This can be an advantage when unbounded outputs
would be problematic, however can lead to problems where an ability to produce out-of-training solutions is necessary.
Additionally, current DT implementations require all nodes (of all trees in the case of RFs and GBDTs) to be held in
memory at inference time, making them potentially memory heavy.
**2.3. Methodologies for Machine Learning**
It is challenging to provide simplified advice for how to approach problem-solving in ML. There are few strict
theoretical reasons to choose any one of the variety of architectures which are available. The authors would also
caution against assuming that results in the literature are the product of a detailed comparison of alternative

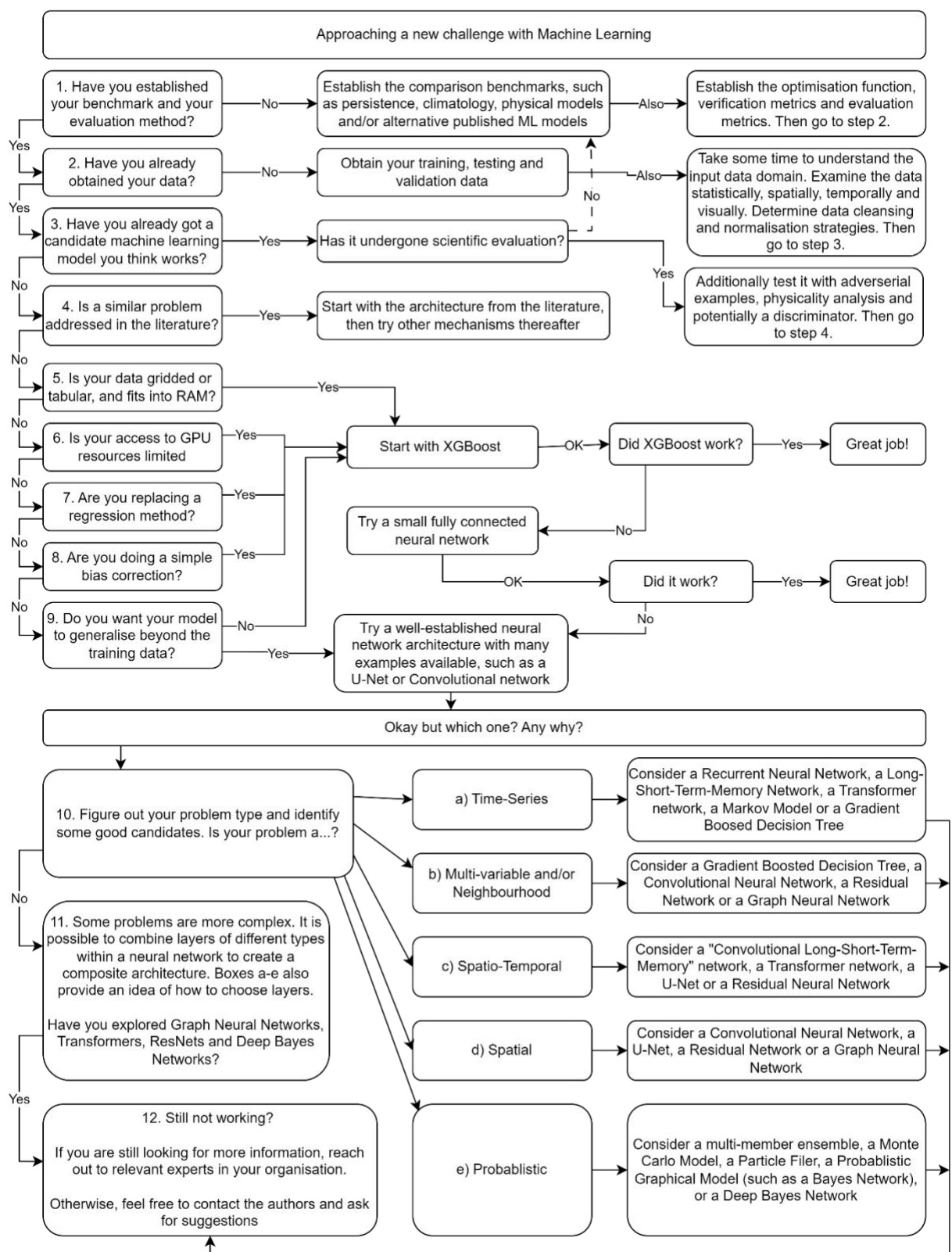

Figure 2: A methodological flowchart illustrating a suggested approach to applying ML to a research problem.
architectures, or assuming that a deep learning approach is going to be easy or straightforward. It will often be the
case that multiple machine learning architectures may be similarly effective, and determining the optimal
architecture is likely to involve extensive iteration. Any specific methodology is also likely to reflect the intuitions
(or biases), knowledge, and background of the authors of that methodology.
Nonetheless, there is an appetite from many scientists for reasonable ways to 'get started' and to provide some
assistance for practical decision-making, particularly if approaching the utilization of machine learning for the first
time or in a new way. Figure 2 provides a set of suggested steps and decision points to help readers approach a new
challenge with ML.
The flowchart presented in Figure 2 provides an overview of methodological steps that can be taken when using ML
to solve a problem, however it does not give much insight into the pros and cons of the common ML architectures
available and used in the literature. Table 1 provides a brief summary of the major ML architectures and algorithms
used by the studies cited in this review and gives a short note on some of their pros and cons. This table is not
exhaustive, and readers are strongly encouraged to use it as a starting point for further exploration, rather than a
definitive guide. The relative strengths and weakness of each ML architecture can be subtle, and highly dependent on
the use case, their application, and their tuning. Establishing a good understanding of the ML architecture being used
is a critical step for any scientist intending to delve into ML modelling. Interested readers should also refer to Chase
et all (2022b), where a similar table is presented that covers a wider variety of traditional methods but fewer neural
network approaches.
An increasingly diverse array ML architectures are being applied to an ever-growing variety of challenges. These
architectures all have sub-variants and ancestor architectures which may not be represented, all of which may be found
to be of use for weather and climate modelling applications. Other concerns, such as data normalization, training
strategies, and capturing physicality become as relevant as the choice of architecture once a certain level of
performance is achieved.
Figure 3 shows a summary of the ML architectures and algorithms used by the studies cited in this review, including
the number of times each architecture is used. It can be seen from this that the two most frequently used general
categories of architecture are Fully Connected NNs (FCNNs) and Convolutional NNs (CNNs) of various sub-types.
However, some of the most significant recent research findings come from new architectures which by definition
cannot have wide adoption yet (these are grouped under the 'Mixed/custom NN' category in Figure 3).
In some cases, little justification is given for the ML architecture used in a study, and readers are therefore cautioned
against using the relative popularity of a particular ML architecture in the literature as a guide for its suitability for a
given task.
Furthermore, ML models increasingly use a mix of different algorithms and architectures. For example, a common
combination is fully-connected NN layers, convolutional NN layers, and LSTM layers. For the purposes of Figure 3,
the authors have endeavoured to categorise the ML architectures used in the studies in this review as accurately as
possible, with complex architectures being placed in the "Mixed/custom NN" category, however, where an
architecture was mostly but not entirely aligned with a single category, it was placed in that category. For example,

| Approach | Description | Pros | Cons |
|---|---|---|---|
| Simple regression techniques | Includes linear regression and logistic regression. See Chase et al. (2022b) for more detail. | Explainable and well-understood. | Can only capture simple relationships. |
| Decision Tree | Consists of a series of branching decisions, culminating is a number of decision 'leaves'. The decision points are trainable.<br>Provides the basis for understanding more complex decision tree and regression tree approaches. | Easily explainable. Computationally tractable and fast | Unable to fully model complex problems. Cannot make predictions outside the training envelope. |
| Random Forest (RF) | A random forest consists of many decision trees, which form an ensemble and the average result is taken. The construction of the trees uses randomness. | Versatile and effective. Computationally tractable and fast. Allows focus on the input variables rather than on process or model definition. | Usually performs slightly less well than gradient boosted decision trees. |
| Gradient Boosted Decision Trees (GBDT) | Akin to Random Forecasts, however each additional member is used to predict the residual error of the ensemble so far.<br>Is often sufficient for a given problem, and should thus be considered as a baseline for measuring more complex ML models against. | A highly versatile and reliable approach. Computationally tractable and fast. Allows focus on the input variables rather than on process or model definition. Feature importance plots can guide intuition. | Has practical limitations at scale due to large memory requirements at inference time. Limited ability to simulate complex systems compared to other ML approaches such as NNs. Cannot make predictions outside the training envelope without customized leaves. |

| Vector Machines | Support Vector Machines (SVMs) and Relevance Vector Machines (RVMs) are supervised models used for regression and classification. RVMs have the same functional form as SVMs, but are a probabilistic classification based on Bayesian inference. Vector Machines seek to define the optimal division between classes by finding the hyperplanes which have the largest distance to the nearest training-data point of any class. | Can be used for similar problems as GBDTs. Computationally efficient and often effective. Mathematically appealing. Capable of modelling nonlinear functions. | Now less-used compared to random forests and GBDTs. |
|---|---|---|---|
| Single neuron | See Chase et al. (2022b) for a description of the structure of a perceptron. Forms the conceptual and structural basis for all NN architectures. | Unused in practice outside of a larger NN architecture. | Unable to model most problems in isolation. |
| Fully-Connected feed-forward Neural Network (FCNN) | Consists of multiple layers of neurons, with each neuron being connected to every neuron in the subsequent layer. Still quite widely used in weather and climate modelling, in spite of declining use in other machine learning domains. Is often sufficient and should be considered as a baseline for measuring more complex architectures against. | Effective for applications such as parametrization scheme emulation and PDE solver preconditioning. Relatively simple to work with. Computationally tractable. | Unable to effectively train beyond a certain size or depth, and thus is increasingly being replaced with more complex architectures as ML moves to deeper NNs. |

| Bayesian networks | A system (probabilistic graphical model) comprised of nodes which together predict both an expected value and a likelihood. Each node is associated with a probability function that provides a probability (or distribution) of the variable represented by the node. | Effective for refining an expert or knowledge-based model by incorporating additional observations. Capable of dealing with both semantic concepts and physical processes. | Determining an optimal model can be challenging and training times are prohibitive for large networks. |
|---|---|---|---|
| Deep Bayesian Networks | Deep Bayesian techniques attempt to capture the model complexity of deep neural networks while retaining the ability to predict a distribution of outcomes, a probabilistic model and a clear information-theoretical bases. | Used to obtain a more realistic expression of uncertainty. Effective in modelling where causal relationships aren't understood. | Not as well explored as neural networks in recent literature. |
| Convolutional Neural Network (CNN) | Involves convolving a (usually 2D image, but can also be 1D temporal, for example) input field with a filter function (often a top hat function[†]) to extract features on different spatial scales. Conceptually useful in understanding how a neural network can build up an abstract or 'big picture' definition of a process in its hidden layers by assembling fine-scale features. | The go-to network for image-based problems. Proven effective on many problems and is well-covered in the literature. | May require more significant hardware such as a modern GPU. |
| Residual Neural Network (ResNet) | ResNets are a form of CNN including skip connections, whereby the inputs of a number of convolutional layers are appended to the outputs of those layers to retain information lost through the weights in the convolutional layers. These skip connections make it possible to train much deeper convolutional networks than would be possible otherwise. | Allows very deep networks to be efficiently trained. Allows an iterative build-up of network size by experimenting with the number of residual layers. Could be a good choice to couple with physically interpretable layers. | Somewhat more computationally costly than other deep architectures. |

| U-Net | Derives its name from the shape of the network as it is commonly shown diagrammatically (it forms a "U" shape).<br><br>Consists of a series of downsampling convolutional layers, each of which further abstracts the information in the inputs (forming the first half of the "U"). These are then upsampled again to the original resolution of the input data (forming the second half of the "U"). Each downsampling step has its output appended to the input of the corresponding upsampling step (a form of skip connection). | Effective for many purposes and widely used in classification and image segmentation. Has also seen uptake for nowcasting applications and prediction of multiyear timescale ocean variables. | No serious drawbacks. Has somewhat given way to more complex architectures recently |
|---|---|---|---|
| Deep Operator Network (DeepONet) | A NN which is designed to learn the mappings between inputs and outputs of the mathematical operators underpinning processes, rather than directly predicting the outputs of the processes themselves. Was developed in the context of fluid dynamics and differential operators.<br><br>An important theoretical component of the Adaptive Fourier Neural Operator used in FourCastNet (Pathak et al., 2022). | Provides a strong theoretical basis for learning the underlying function space of a data set.<br><br>Highly effective for fluid dynamics and idealized systems.<br><br>Can retain the properties of the learned operators. For example, can exhibit translational and scale invariance where that property holds for the operator in question. | Conceptually not straightforward. Requires strong mathematical and machine learning expertise to apply effectively to new challenges. |

| Graph Neural Network (GNN) | Models data as a set of interconnected nodes and edges (as opposed to assuming data is on a regular grid). Underpins Keisler (2022) and GraphCast (Lam et al., 2022) | Does not require data to be on a grid or distributed in a uniform manner. Capable of incorporating teleconnections, nonlocal relationships, and other complex variable relationships. | Costly to train. |
|---|---|---|---|
| Discriminator | A NN is trained to discriminate between two examples and identify the "real" one. Is used to estimate whether a sample is from the observations or the model. Forms one part of a GAN. | Can be used in place of a manually-defined loss function to train without over-emphasizing any individual metrics or variables. Can be used as an effective loss function when training Can be used independently to evaluate model realism. Comes closest to human subjective evaluation of image quality. | Is more likely to require more machine learning domain knowledge to resolve issues. |
| Generative Adversarial Network (GAN) | Combines a generator network with a discriminator and trains them in an adversarial manner: the discriminator tries to differentiate the generator from ground truth, the generator tries to trick the discriminator. Eventually the discriminator can't differentiate the generator from ground truth. May be part of a multi-phase training strategy in order to improve realism after initial optimization. | Produce results which prioritize realism over accuracy (could also be a con). Is less prone to the blurring that results from training to simpler loss functions and thus can be more effective in producing sharp images and predicting statistical extremes. | Increases training costs. Favors a 'good looking' answer over a correct answer. Can be difficult to train as the generator and discriminator must be kept balanced (one can outperform the other leading to mode collapse – a false minima). |

| | | | |
|---|---|---|---|
| Recurrent Neural Network (RNN) | Any neural network where the output of previous predictions are provided to a sequence-based model. Multiple sub-types of the RNN exist. | A simple RNN design can model many problems effectively. A recurrent architecture allows access to and inspection of the belief state at each iteration. | Recurrent approaches can accumulate errors quickly. Relationships which act over longer time-frames or distances than the recurrence length may not be captured. Choosing the length of the sequence may be a challenge. |
| Long Short Term Memory (LSTM) Network | Contains modified neurons with a memory component and the ability to retain or forget information. Is applied to sequence inputs and can learn the sequential scales in which information is encoded (e.g., what timescales in a timeseries are pertinent for future prediction). Has been combined with the ideas underpinning CNNs to create Convolutional LSTMs (ConvLSTM), which fit for both timescales of relevance and spatial features of relevance. | An effective alternative to a recurrent network which has proven very good at modelling time-series. A proven and effective mechanism for dimensionality reduction to allow the training of large networks. | May not include spatial relationships (unless it's a ConvLSTM), and may be more complex than needed for some problems. Less explainable than an attention mechanism. Has a bias towards closer points in a sequence (e.g., will be biased towards the recent past over a longer timescale in time series prediction). |

| | | | |
|---|---|---|---|
| Attention Mechanism | Often used in conjunction with other architectures as a feature extraction/dimensionality reduction method.<br><br>A NN is trained to learn the degree of importance of each input datapoint on each other one in a sequence.<br><br>Attention mechanism-based NNs are rapidly overtaking LSTMs as the method of choice for modelling sequence-based information. | Unlike LSTMS, attention mechanisms are not biased towards relationships between near points in a sequence. Rather, attention mechanisms treat all points in an input sequence equally and retain the learned attention mappings between each point.<br><br>In the context of weather and climate modelling, the learned attention mappings between points can be a useful tool for assessing the degree to which a NN has learned physically realistic teleconnections. | More costly to train than an LSTM for the same problem because attention mechanisms have more free parameters. |
| Transformer | The transformer architecture combines an attention mechanism with an autoregressive approach whereby each previously predicted step in a sequence is an input into the prediction of the next step.<br><br>Transformer architectures underpin the current generation of language models such as ChatGPT.<br><br>Transformers are now often included as part of other architectures for input dimensionality reduction. | A proven and effective mechanism for dimensionality reduction to allow the training of large networks.<br><br>While the uptake of transformer architectures in weather and climate modelling is still small, their impressive performance for sequence prediction suggests they could have great for the field. | Transformers can be difficult to train due to a tendency to overemphasize the recurrent component of the network over new inputs in the early stages of training. |


Table 1: A summary of major ML architectures and algorithms used by the studies cited in this review. Interested readers should
also refer Chase et all (2022b) where a similar table is presented that covers a wider variety of traditional methods but fewer neural
network approaches.

an LSTM model with a small number of feed-forward layers would be categorised as a Recurrent NN. Since many
contemporary ML models combine multiple architectural elements and algorithms into the one model, it is somewhat
of an oversimplification to consider each of these in isolation, and while starting with a simple model design with a
limited selection of layer types is advisable to aid interpretability, there is no reason they cannot be combined or used
in conjunction with each other if this improves the performance of the model.
Adapting, optimizing and debugging issues with machine learning systems can be very complex (especially so for
large NNs), and is likely to require both machine learning expertise and domain knowledge (i.e. scientific knowledge).
XGBoost provides the ability to generate chart showing the importance of the features in the model which can be very
helpful. Shapley Additive Explanations (Lundberg and Lee 2017) can provide insights into feature importance for any
model including NNs.

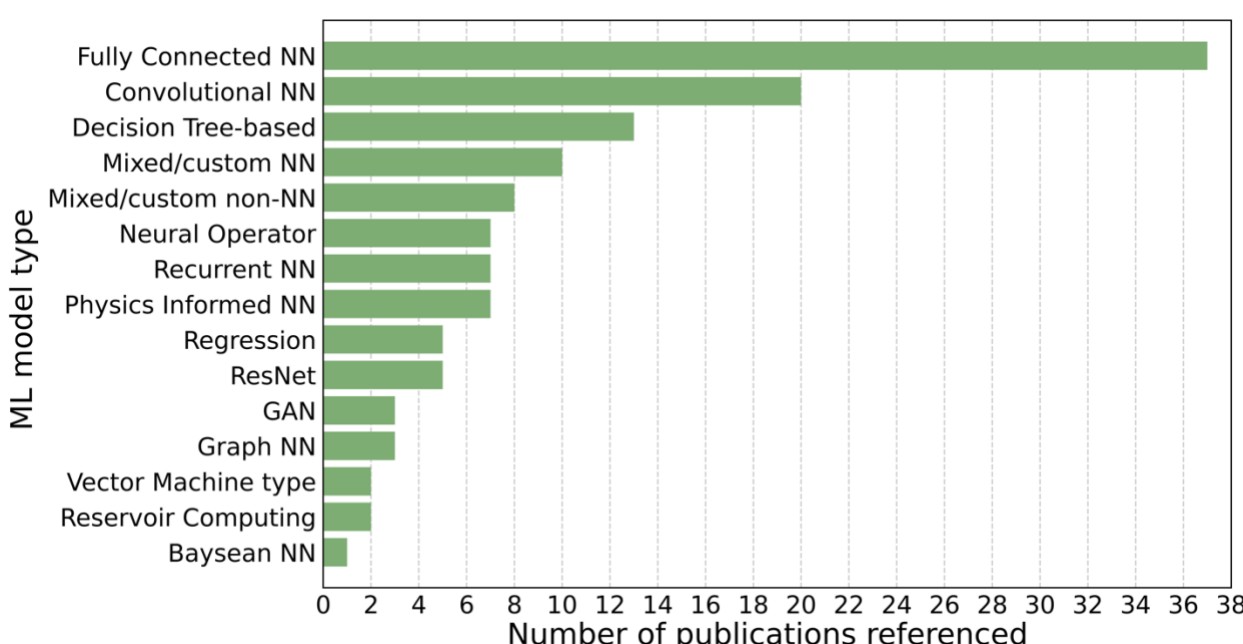


Figure 3: A count of the ML architectures and algorithms used by the studies cited in this review. As with Figure 1, this figure
includes all references from this review except for: seminal ML papers that are on new ML methods (e.g., foundational ML papers),
review papers, any paper cited that concerns a topic which is out of scope (e.g., nowcasting), and any other paper which does not
present a new method directly applicable to weather and climate modelling. The full table of citations is provided in the appendix.

**3. Sub-grid parametrization and emulation**
Subgrid-scale processes in numerical weather and climate models are typically represented via a statistical
parameterization of what the macroscopic impacts of the process would be on resolved processes and parameters.
These are commonly referred to as parameterization schemes, and can be very complex and relatively computationally
costly. For example, in the European Centre for Medium-Range Weather Forecast's (ECMWF) Integrated Forecasting
System (IFS) model they account for about a third of the total computational cost of running the model (Chantry et al.
2021b). They also require some understanding of the underlying unresolved physical processes. Examples of subgrid-
scale processes which are typically currently parameterized in operational systems include gravity wave drag,
convection, radiation, subgrid-scale turbulence, and cloud microphysics. As additional complexity (for example
representation of aerosols, atmospheric chemistry, land surface processes, etc.) is added to numerical models, the
computational cost will only increase.
ML presents an alternative approach to representing subgrid-scale processes, either by emulating the behavior of an
existing parametrization scheme, emulating the behavior of sub-components of the scheme, by replacing the current
scheme or sub-component entirely with an ML-based scheme, or by replacing the aggregate effects of multiple
parametrization schemes with a single ML model.
ML emulation of existing schemes or sub-components has the advantage of maintaining the status quo within the
model; no or minimal re-tuning of the model should be required since the ML emulation is trained to replicate the
results of an already-tuned-for scheme. Because of this, the main benefit of this approach is that it reduces the
computational cost of running the parametrization scheme. On the other hand, full replacement of an existing
parameterization scheme or sub-component with an ML alternative has the potential to be both computationally
cheaper and also an improvement over the preceding scheme.
In the following subsections, a review of the literature on aspects of ML for the parametrization and emulation of
subgrid-scale processes is presented.

### 3.1. Early work on ML parametrization and ML emulations

A popular target for applying ML in climate models is radiative transfer, since it is one of the more computationally
costly components of the model. As such, many early examples of the use of ML in sub-grid parametrization schemes
focus on aspects of this physical process. Chevallier et al. (1998) trained NNs to represent the radiative transfer budget
from the top of the atmosphere to the land surface, with a focus on application in climate studies. They incorporated
the information from both line-by-line and band models in their training to achieve competitive results against both
benchmarks. Their NNs achieved accuracies comparable to or better than benchmark radiative transfer models of the
time, while also being much faster computationally.
In contrast to the ML based scheme developed by Chevallier et al. (1998), which could be considered an entirely new
parametrization scheme, Krasnopolsky et al. (2005) used NNs to develop an ML based emulation of the existing
atmospheric longwave radiation parametrization scheme in the NCAR Community Atmospheric Model (CAM). The
authors demonstrated speedups with the NN emulation of 50-80 times the original parameterization scheme.
Emulation of existing schemes has since then become a popular method for achieving significant model speedups. For
example, Gettelman et al. (2021) investigated the differences between a General Circulation Model (GCM) with the
warm rain formation process replaced with a bin microphysical model (resulting in a 400% slowdown) and one with
the standard bulk microphysics parameterization in place. They then replaced the bin microphysical model with a set
of NNs designed to emulate the differences observed, and showed that this configuration was able to closely reproduce
the effects of including the bin microphysical model, without any of the corresponding slowdown in the GCM.

**3.2. ML for coarse graining**

Coarse graining involves using higher resolution model or analysis data to map the relationship between smaller scale processes and a coarser grid resolution. It can be used to develop parameterization schemes without explicitly representing the physics of smaller scale processes.

This has proven to be a popular method for developing ML-based parametrization schemes. Brenowitz & Bretherton (2018) used a near-global aqua planet simulation run at 4 km grid length to train a NN to represent the apparent sources of heat and moisture averaged onto 160 $km^2$ grid boxes. They then tested this scheme in a prognostic single column model and showed that it performed better than a traditional model in matching the behavior of the aqua planet simulation it was trained on. Brenowitz & Bretherton (2019) built on this work by training their NN on the same global aqua-planet 4 km simulation, but then embedded this scheme within a coarser resolution (160 $km^2$) global aqua planet GCM. Embedding NNs within GCMs is challenging because feedbacks between NN and GCM components can cause spatially extended simulations to become dynamically unstable within a few model days. This is due to the inherently chaotic nature of the atmosphere in the GCM responding to inputs from the NN which cause rapidly escalating dynamical instabilities and/or violate physical conservation laws. The authors overcame this by identifying and removing inputs into the NN which were contributing to feedbacks between the NN and GCM (Brenowitz et al. 2020), and by including multiple time steps in the NN training cost function. This resulted in stable simulations which predicted the future state more accurately than the course resolution GCM without any parametrization of subgrid-scale variability, however the authors do observe that the mean state of their NN-coupled GCM would drift, making it unsuitable for prognostic climate simulations.

Rasp et al. (2018) trained a deep NN[†] to represent all atmospheric subgrid processes in an aquaplanet climate model by learning from a multiscale model in which convection was treated explicitly. They then replaced all sub-grid parameterizations in an aquaplanet GCM with the deep NN, and allowed it to freely interact with the resolved dynamics and the surface-flux scheme. They showed that the resulting system was stable and able to closely reproduce not only the mean climate of the cloud-resolving simulation but also key aspects of variability in prognostic multiyear simulations. The authors noted that their decision to use deep NNs was a deliberate one, because they proved more stable in their prognostic simulations than shallower NNs, and they also observed that larger networks achieved lower training losses. However, while Rasp et al. (2018) were able to engineer a stable model that produced results close to the reference GCM, small changes in the training dataset or input and output vectors quickly led to the NN producing increasingly unrealistic outputs and causing model blow-ups (Rasp 2020). Consistent with this, Brenowitz & Bretherton (2019) report that they were unable to achieve the same improvements in stability with increasing network layers found by Rasp et al. (2018).

**3.3. Overcoming instability in ML emulations and parametrizations**

O'Gorman & Dwyer (2018) tackled the instabilities observed in NN-based approaches to subgrid-scale parameterization by employing an alternative ML method; Random Forests (RFs; Breiman 2001; Tibshirani & Friedman 2001). The authors trained a RF to emulate the outputs of a conventional moist convection parametrization

scheme. They then replaced the conventional parameterization scheme with this emulation within a global climate
model, and showed that it ran stably and was able to accurately produce climate statistics such as precipitation
extremes without needing to be specially trained on extreme scenarios. RFs consist of an ensemble of DTs, with the
predictions of the RF being the average of the predictions of the DTs which in turn exist within the domain of the
training data. RFs thus have the property that their predictions cannot go outside of the domain for their training data,
which in the case of O'Gorman & Dwyer (2018) ensured conservation of energy and nonnegativity of surface
precipitation (both critically important features of the moist convection parametrization scheme) were automatically
achieved. A disadvantage of this method however is that it requires considerable memory when the climate model is
being run to store the tree structures and predicted values which make up the RF.
Yuval & O'Gorman (2020) extended on the ideas in O'Gorman & Dwyer (2018), switching from emulation of a single
parametrization scheme to emulation of all atmospheric sub grid processes. They trained an RF on a high-resolution
three-dimensional model of a quasi-global atmosphere to produce outputs for a course-grained version of the model,
and showed that at course resolution the RF can be used to reproduce the climate of the high-resolution simulation,
running stably for 1000 days.
There are some drawbacks to a RF approach compared to a NN approach however; namely that NNs may provide the
possibility for greater accuracy than RFs, and also require substantially less memory when implemented. Given that
GCMs are already memory intensive this can be a limiting factor in the practical application of ML parametrization
schemes. Furthermore, there is the potential to implement reduced precision NNs on Graphics Processing Units
(GPUs) and Central Processing Units (CPUs) which still achieve sufficient accuracy, leading to substantial gains in
computational efficiency. Motivated by these considerations, Yuval et al. (2021) trained a NN in a similar manner to
how the RF in Yuval & O'Gorman (2020) was trained, using a high resolution aqua-planet model and aiming to coarse
grain the model parameters. They overcame the model instabilities observed to occur in previous attempts to use NNs
for this process by wherever possible training to predict fluxes and sources and sinks (as opposed to the net tendencies
predicted by the RF in Yuval & O'Gorman (2020)), thus incorporating physical constraints into the NN
parametrization. The authors also investigated the impact of reduced precision in the NN, and found that it had little
impact on the simulated climate.
**3.4. From aquaplanets to realistic land-ocean simulations**
All of the studies discussed in this section so far which were tested in a full GCM have used aqua planet simulations.
Han et al. (2020) broke away from this trend by developing a Residual NN[†] (ResNet) based parametrization scheme
which emulated the moist physics processes in a realistic land-ocean simulation. Their emulation reproduced the
characteristics of the land-ocean simulation well, and was also stable when embedded in single column models.
Mooers et al. (2021) represents a subsequent example of an ML emulation of atmospheric fields with realistic
geographical boundary conditions, where the authors developed feed-forward NNs to super-parametrize subgrid-scale
atmospheric parameters and forced a realistic land surface model with them. Super-parametrization is distinct from
traditional parameterization in that it relies on solving (usually simplified) governing equations for subgrid-scale
processes rather than heuristic approximations of these processes. They employed automated hyperparameter
optimization[†] to investigate a range of neural network architectures across ~250 trials, and investigated the statistical
characteristics of their emulations. While the authors found that their NNs had a less good fit in the tropical marine
boundary layer, attributable to the NN struggling to emulate fast stochastic signals in convection, they also reported
good skill for signals on diurnal to synoptic timescales.
Brenowitz et al. (2022) sought to address the challenge of emulating fast processes. They used FV3GFS (Zhou et al.,
2019; Harris et al., 2021; a compressible atmospheric model used for operational weather forecasts by the US National
Weather Service) with a simple cloud microphysics scheme included to generate training data and used this to train a
selection of ML models to emulate cloud microphysics processes, including fast phase changes. They emulated
different aspects of the microphysics with separate ML models chosen to be suitable to each task. For example, simple
parameters were trained with single-layer NNs, while parameters which are more complex spatially were trained with
RNNs (e.g., rain falls downwards and not upwards, so it is sequential in timesteps through the atmosphere – a feature
which can be represented by an RNN). They then embedded their ML emulation in FV3GFS. They found that their
combined ML simulation performed skillfully according to their chosen metrics, but had excessive cloud over the
Antarctic Plateau.
All of these studies, however, did not test their parameterizations in prognostic long-term simulations.
**3.5. Testing with prognostic long-term simulations**
A barrier to achieving stable runs with minimal model drift with ML components is the fact that generic ML models
are not designed to conserve quantities which are required to be conserved by the physics of the atmosphere and ocean.
Beucler et al. (2019) proposed and tested two methods for imposing such constraints in a NN model; (1) constraining
the loss function or (2) constraining the architecture of the network itself. They found that their control NN with no
physical constraints imposed performed well, but did so by breaking conservation laws, bringing into question the
trustworthiness of such a model in a prognostic setting. Their constrained networks did however generalize better to
unforeseen conditions, implying they might perform better under a changing climate than unconstrained models.
Chantry et al. (2021b) trained a NN to emulate the non-orographic gravity wave drag parameterization in the ECMWF
IFS model (specifically cycle 45R1, ECMWF, 2018) and were able to run stable, accurate simulations out to 1 year
with this emulation coupled to the IFS. While the authors note that RFs have been shown to be more stable (e.g.,
O'Gorman & Dwyer (2018) and Yuval & O'Gorman (2020), as described above, and Brenowitz et al. (2020)), they
chose to focus on NNs since they have lower memory requirements and therefore promise better theoretical
performance. The authors assessed the performance of their emulation in a realistic GCM by coupling the NN with
the IFS, replacing the existing non-orographic gravity wave drag scheme, and performed 120 hour, 10 day, and 1 year
forecasts at ~25 km resolution in a variety of model configurations. The authors showed that their emulation was able
to run stably when coupled to the IFS for seasonal timescales, including being able to reproduce the descent of the
Quasi-biennial Oscillation (QBO). Interestingly, while the authors initially aimed to ensure momentum conservation
in a manner similar to Beucler at al. (2021), they found that this constraint led to model instabilities and that a better

result was achieved without it. One possible explanation for this is that Beucler at al. (2021) assessed their NNs in an aquaplanet setting. Nonetheless, Chantry et al. (2021b) noted that since their method was not identical to Beucler et al. (2021), improved stability could potentially be achieved by following their method more precisely. The computational cost of the NN emulation developed by Chantry et al. (2021b) was found to be similar that of the existing parametrization scheme when run on CPUs, but was faster by a factor of 10 when run on GPUs due to the reduction in data transmission bottlenecks.

The first study to successfully run stable long-term climate simulations with ML parametrizations was Wang et al. (2022a), who extended on the work of Han et al. (2020) by constructing a ReNet to emulate moist physics processes. They used the residual connections from Han et al. (2020) to construct NNs with good nonlinear fitting ability, and filtered out unstable NN parametrizations using a trial-and-error analysis, resulting in the best ResNet set in terms of accuracy and long-term stability. They implemented this scheme in a GCM with realistic geographical boundary conditions and were able to maintain stable simulations for over 10 years in an Atmospheric Model Intercomparison Project (AMIP)-style configuration. This was more akin to a hybrid ML-physics based model than a traditional GCM with ML-based parametrization, because rather than embedding the ResNet in the model code, the authors used a NN-GCM coupling platform through which the NNs and GCMs could interact through data transmission. This is in contrast to the approach employed in the Physical-model Integration with Machine Learning[4] (PIML) project and Infero[5], which are both described in Section 3.11. One advantage to this approach noted by the authors is that it allows for a high degree of flexibility in the application of the ML component, however is likely to be less efficient than a fully-embedded ML model, due to the potential for data transmission bottlenecks.

**3.6. Training with observational data**

An alternative to using more complex and/or higher resolution models for training data is to train using direct observational data. For example, Ukkonen & Mäkelä (2019) used reanalysis data from ERA5 and lightning observation data to train a variety of different types of ML models to predict thunderstorm occurrence; this was then used as a proxy to trigger deep convection. ML models assessed were logistic regression, RFs, GBDTs, and NNs, with the final two showing a significant increase in skill over convective available potential energy (CAPE; a standard measure of potential convective instability). One of the challenges of accurately reproducing the large-scale effects of convection is correctly identifying when deep convection should occur within a grid cell. The authors proposed that an ML model such as those they assessed could be used as the "trigger function" which activates the deep convection scheme within a GCM.

**3.7. ML for super parameterization**

Revisiting the topic of super parametrized subgrid-scale processes introduced above, the use of ML for this approach was investigated in depth by Chattopadhyay et al. (2020). The authors introduced a framework for NN-based super

---

[4][https://turbo-adventure-f9826cb3.pages.github.io](https://turbo-adventure-f9826cb3.pages.github.io) accessed 7th February 2023
[5]https://infero.readthedocs.io/en/latest/ accessed 7th February 2023

parametrization, and compared the performance of this method against NN-based traditional parametrization (i.e.,
based on heuristic approximations of subgrid-scale processes) and direct super parameterization (i.e., explicitly
solving for the subgrid-scale processes) in a chaotic Lorenz '96 (Lorenz 1996) system that had three sets of variables,
each of a different scale. They found that their NN-based super parameterization outperformed direct super
parameterization in terms of computational cost, and was more accurate than NN-based traditional parametrization.
The NN-based super parameterization showed comparable accuracy to direct super parameterization in reproducing
long-term climate statistics, but was not always comparable for short-term forecasting.
**3.8. Stochastic parametrization schemes**
A more recent approach to the representation of subgrid-scale processes is via stochastic parameterization schemes,
which can represent uncertainty within the scheme. There has been less focus on replacing these schemes with ML
alternatives than non-stochastic schemes, however some progress has been made. Krasnopolsky et al. (2013) used an
ensemble of NNs to learn a stochastic convection parametrization from data from a high-resolution cloud resolving
model. In this case, the stochastic nature of the parametrization was captured by the ensemble of NNs. Gagne et al
(2020b) took a different approach, investigating the utility of generative adversarial networks (GANs) for stochastic
parametrization schemes in Lorenz '96 (Lorenz 1996) models. In this case, the GAN learned to emulate the noise of
the scheme directly, rather than implicitly representing it with an ensemble. They described the effects of different
methods to characterize input noise for the GAN, and the performance of the model at both weather and climate
timescales. The authors found that the properties of the noise influenced the efficacy of training. Too much noise
resulted impaired model convergence and too little noise resulted in instabilities within the trained networks.
**3.9. ML parametrization and emulation for land, ocean, and sea ice models**
Models of the atmosphere make up one component of the Earth system, however for timescales beyond a few days,
simulating other components of the Earth system becomes increasingly important to maintain accuracy. The
components which are most often included in coupled Earth system models in addition to the atmosphere are the
ocean, sea ice, and the land surface. Reflective of this, ML approaches to parameterization of subgrid-scale processes
are not limited to the atmosphere, and progress has been made in the use of ML for land, ocean and sea ice models as
well.
On the ocean modelling front, Krasnopolsky et al. (2002) presented an early application of NN for the approximation
of seawater density, the inversion of the seawater equation of state, and a NN approximation of the nonlinear wave-
wave interaction. More recently, Bolton & Zanna (2019) investigated the utility of Convolutional Neural Networks
(CNNs) for parametrizing unresolved turbulent ocean processes and subsurface flow fields. Zanna & Bolton (2020)
then investigated both Relevance Vector Machines[†] (RVMs) and CNNs for parameterizing mesoscale ocean eddies.
They demonstrated that because RVMs are interpretable, they can be used to reveal closed-form equations for eddy
parameterizations with embedded conservation laws. The authors tested the RVM and CNN parameterizations in an
idealized ocean model and found that both improved the statistics of the coarse resolution simulation. While the CNN

was found to be more stable than the RVM, the advantage of the RVM was the greater interpretability of its outputs. Finally, Ross et al. (2023) developed a framework for benchmarking ML based parametrization schemes for subgrid-scale ocean processes. They used CNNs, symbolic regression, and genetic programming methods to emulate a variety of subgrid-scale forcings including measures of potential vorticity and velocity, and developed a standard set of metrics to evaluate these emulations. They found that their CNNs were stable and performed well when implemented online, but generalized poorly to new regimes.

Focusing instead on sea ice, Chi & Kim (2017) assessed the ability of two NN models; a fully connected NN and an LSTM, to predict Antarctic sea ice concentration up to a year in advance. Their ML models outperformed an autoregressive model comparator, and were in good agreement with observed sea ice extent. Andersson et al. (2021) improved upon this work with their model IceNet, A U-Net ensemble model which produced probabilistic Arctic sea ice concentration predictions to a 6-month lead time. The authors compared IceNet to the SEAS5 dynamical sea ice model (Johnson et al., 2019) and showed an improvement in the accuracy of a binary classification of ice/no ice for all lead months except the first month. Horvat & Roach (2022) used ML to emulate a parameterization of wave-induced sea ice floe fracture they had developed previously, in order to reduce the computational cost of the scheme. When embedded in a climate simulation, their ML scheme resulted in an overall categorical accuracy (accounting for the fact that it was only called where needed) of 96.5%. However, the authors did note that since their ML scheme was trained on present day sea ice conditions, it may have reduced success under different climate scenarios, and they recommend retraining using climate model sea-ice conditions to account for this. Rosier et al. (2023) developed MELTNET, a ML emulation of the ocean induced ice shelf melt rates in the NEMO ocean model (Gurvan et al., 2019). MELTNET consisted of a melt rate segmentation task, followed by a denoising autoencoder network which converted the discrete labelled melt rates to a continuous melt rate. The authors demonstrated that MELTNET generalized well to ice shelf geometries outside the training set, and outperformed two intermediate-complexity melt rate parameterizations, even when parameters in those models were tuned to minimize any misfit for the geometries used. Given the computational cost of sea ice parametrizations is relatively high for the timescales on which sea ice evolution is important (namely, seasonal to climate timescales), and given the promising results in emulating these parametrizations demonstrated in the literature, ML based emulation of these schemes is a strong candidate for inclusion into future dynamical coupled modelling systems.

Finally, considering Earth's surface, most of the focus of ML innovations in this context has focused on land use classification (e.g, Carranza-García et al, 2019; Digra et al., 2022) and crop modelling (e.g., Virnodkar et al., 2020; Zhang et al., 2023). The rate of publication of ML applications for land surface models has been slower, however there has nonetheless been steady progress in this space in recent years. Pal & Sharma (2021) presented a review of the use of ML in land surface modelling which provides an excellent primer of the state of the field to that point. They include in their review an overview of land surface modelling components and processes, before reviewing the literature on the use of ML to represent them. They separate their review into attempts to predict and parametrize different variables or aspects of the model, including evapotranspiration (Alemohammad et al., 2017; Zhao et al., 2019; Pan et al., 2020), soil moisture (Pelissier et al., 2020), momentum and heat fluxes (Leufen & Schädler, 2019), and parameter estimation and uncertainty (Chaney et al., 2016; Sawada, 2020; Dagon et al., 2020). They also provide

a useful summary of the ML architectures that have been used in publications they have discussed. More recently, He
et al. (2022) developed a hybrid approach to modelling aspects of the land surface, where a traditional land surface
model was used to optimize selected vegetation characteristics, while a coupled ML model simulated a corresponding
three-layer soil moisture field. The estimated evapotranspiration from this hybrid model was compared to observations
and it was found that it performed well in vegetated areas but underestimated the evapotranspiration in extreme arid
deserts. The ready application of ML to aspects of land surface modelling, and the relative sparsity of publications in
this space suggests that it is a fertile domain for further research and development.
**3.10. ML for representing or correcting a sub-component of a parametrization scheme**
An alternative method to replacing or emulating an entire parametrization scheme or schemes with ML is to target the
most costly or troublesome sub-components of the scheme, and either replace those or make corrections to them.
Ukkonen et al. (2020) trained NNs to replace gas optics computations in the RTE-RRTMGP (Radiative Transfer for
Energetics and Rapid and accurate Radiative Transfer Model for General circulation models applications-Parallel;
Pincus et al., 2019) scheme. The NNs were faster by a factor of 1-6, depending on the software and hardware platforms
used. The accuracy of the scheme remained similar to that of the original scheme.
Meyer et al. (2022) trained a NN to account for the differences between 1D cloud effects in the European Centre for
Medium Range Weather Forecasting (ECMWF) 1D radiation scheme ecRad and 3D cloud effects in the ECMWF
SPARTACUS (SPeedy Algorithm for Radiative TrAnsfer through CloUd Sides) solver. The 1D cloud effects solver
within ecRad, Tripleclouds, is favored over the 3D SPARTACUS solver because it is five times less computationally
expensive. The authors show that their NN can account for differences between the two schemes with typical errors
between 20% and 30% of the 3D signal, resulting in an improvement in Tripleclouds' accuracy with an increase in
runtime of approximately 1%. By accounting for the differences between SPARTACUS and Tripleclouds rather than
emulating all of SPARTACUS, the authors were able to keep Tripleclouds unchanged within ecRad for cloud-free
areas of the atmosphere, and utilize the NN 3D correction elsewhere.
**3.11. Bridging the gap between popular languages for ML and large numerical models**
A common toolset for researchers to develop and experiment with different ML approaches to problems is Python
libraries such as pytorch, scikit-learn, tensorflow, keras, etc., or other dynamically-typed, non-precompiled languages.
In contrast, numerical weather models are almost universally written in statically-typed compiled languages,
predominantly Fortran. To make use of ML emulations or parameterizations in the models thus requires that they be:
(1) treated as a separate model periodically coupled to the main model (as is done between atmosphere and ocean
models for example), or
(2) be manually re-implemented in Fortran, or
(3) that the pre-existing libraries used are somehow be made accessible within the model code.
Wang et al. (2022a; mentioned already above) opted for method 1, developing what could be considered a hybrid ML-
physics based model rather than a traditional GCM with ML-based parametrization. In their study, the authors used a

NN-GCM coupling platform through which the NNs and GCMs could interact through data transmission. One advantage to this approach noted by the authors is that it allows for a high degree of flexibility in the application of the ML component, however, is likely to be less efficient than a fully-embedded ML model, due to the potential for data transmission bottlenecks. This framework was then formalized by Zhong et al. (2023).

There are many examples where method 2 was used, such as Rasp et al. (2018), Brenowitz & Bretherton (2018), Gagne et al. (2019) and Gagne et al. (2020a). The obvious disadvantage of this approach is that every change to the ML model being used requires reimplementation in the Fortran, and if the aim is to test a suite of ML models, this approach becomes untenable. Furthermore, this approach poses greater technical barriers for scientists developing ML-based solutions for numerical model challenges, since they must be sufficiently proficient in Fortran to reimplement models in it, rather than using existing user-friendly Python toolkits.

A solution lying somewhere between methods 2 and 3 was developed by Ott et al. (2020), who developed a Fortran-Keras Bridge (FKB) library that facilitated the implementation of Keras-like[†] NN modules in Fortran, providing a more modular means to build NNs in Fortran code. This however did not fully overcome the drawbacks posed by method 2 on its own; implementation of layers in the Fortran is still necessary, and any innovations in the Python modules being used would need to be mirrored in the Fortran library.

Finally, method 3 is being tackled by the Met Office in the PIML[6] project, and by ECMWF with an application called Infero[7]. These projects both seek to develop a framework which can be used by researchers to develop ML solutions to modelling problems in Python, and then integrate them directly into the existing codebase of the physical model (e.g., the Unified model at the UK Met Office). The approach used is to directly expose the compiled code underpinning the Python modules within the physical model code.

**4. Application of ML for the partial differential equations governing fluid flow**

The representation and solving of the partial differential equations (PDEs) governing the fluid flow and dynamical processes in the oceans and atmosphere can be considered the backbone of weather and climate models. The solvers used to find solutions to these equations are typically iterative, and must solve the dynamics-governing equations of their model on every timestep and at every grid point. There has been growing interest in using ML to facilitate speedups and computational cost reductions in the preconditioning and execution of these solvers. Preconditioners are used to reduce the number of iterations required for a solver to converge on a solution, and usually do so by inverting parts of the linear problem. Many earlier studies focused on using ML to select the best preconditioner and/or PDE solver from a set of possible choices (e.g. Holloway & Chen, 2007; Kuefler & Chen, 2008; George et al., 2008; Peairs & Chen, 2011; Huang et al., 2016; and Yamada et al., 2018). Ackmann et al. (2020) approached the preconditioner part of the system more directly, using a variety of ML methods to directly predict the pre-condition of a linear solver, rather than using a standard preconditioner. Rizzuti et al. (2019) focused on the solver, using ML to apply corrections

---

[6] https://turbo-adventure-f9826cb3.pages.github.io/ accessed 7th February 2023
[7] https://infero.readthedocs.io/en/latest/ accessed 7th February 2023

to a traditional iterative solver for the Helmholtz equation. Going a step further, a number of studies have used ML to
replace the linear solver entirely (Ladický et al., 2015; Yang et al., 2016; Tompson et al., 2017).
Representation of the fluid equations in a gridded model poses a challenge because of the inability to resolve fine
features in their solution. This leads to the use of course-grained approximations to the actual equations, which aim to
accurately represent longer-wavelength dynamics while properly accounting for unresolved smaller-scale features.
Bar-Sinai et al. (2019) trained a NN to optimally discretize the PDEs based on actual solutions to the known underlying
equations. They showed that their method is highly accurate, allowing them to integrate in time a collection of
nonlinear equations in 1 spatial dimension at resolutions $4\times$ to $8\times$ coarser than was possible with standard finite-
difference methods.
Building on this, Kochkov et al. (2021) developed a ML-based method to accurately calculate the time evolution of
solutions to nonlinear PDEs which used grids an order of magnitude coarser than is traditionally required to achieve
the same degree of accuracy. They used convolutional NNs to discover discretized versions of the equations (as in
Bar-Sinai et al., 2019), and applied this method selectively to the components of traditional solvers most affected by
coarse resolution, with each NN being equation specific. They utilized the property that the dynamics of the PDEs
were localized, combined with the convolutional layers of their NN enforcing translation invariance[†], to perform their
training simulations on small but high-resolution domains, making the training set affordable to produce. An
interesting feature of their training approach, which is growing in popularity, was the inclusion of the numerical solver
in the training loss function: the loss function was defined as the cumulative pointwise error between the predicted
and ground truth values over the training period. In this way, the NN model could see its own outputs as inputs,
ensuring an internally-consistent training process. This had the effect of improving the predictive performance of the
model over longer timescales, in terms of both accuracy and stability. Finally, the authors demonstrated that their
models produced generalizable properties (i.e., although the models were trained on small domains, they produced
accurate simulations over larger domains with different forcing and Reynolds number). They showed that this
generalization property arose from consistent physical constraints being enforced by their chosen method.
An alternative to using ML to discover discretized versions of the PDE equations is to instead use NNs to learn the
evolution operator of the underlying unknown PDE, a method often referred to as a DeepONet[†]. The evolution operator
maps the solution of a PDE forwards in time and completely characterizes the solution evolution of the underlying
unknown PDE. Because it is operating on the PDE, it is scale invariant and so bypasses the restriction of other methods
that must be trained for a specific discretization or grid scale. Interest in, and the degree of sophistication of,
DeepONets has grown rapidly in recent years (e.g., Lu et al., 2019; Wu & Xiu, 2020; Bhattacharya et al., 2020; Li et
al., 2020a; Li et al., 2020b; Li et al., 2020c; Nelsen & Stuart, 2021; Patel et al., 2021; Wang et al., 2021; Lanthaler et
al. 2022), to the point where the method is showing promising speedups: 3x faster than traditional solvers in the case
of Wang et al. (2021).
The application of ML to the solving of PDEs and the preconditioning of PDE solvers has been a fruitful avenue of
research to date. It has led to innovations which have proven useful even outside of the immediate field (e.g., Pathak
et al. 2022 adapted innovations from DeepONets to use in fully ML-based weather models - this is discussed further
in the next Section). This is likely in part because there are many areas of engineering and science which are active in
progressing relevant research, leading to a greater overall pace of innovation. ML-based PDE solvers and
preconditioners have not yet been tested in a physical weather and climate model. There are few theoretical reasons
this could not occur and, if effective, result in significant computational efficiencies for traditional physical model
architectures. This poses an interesting avenue for further research.
**5. Numerical model replacement/emulation**
The shift from using ML to emulate or replace parametrization schemes to using ML to replace the entire GCM has
been made plausible by the increasing volume of training data available. The focus in this section will be on the
challenge of completely replacing a GCM with a ML model.
There has been a flurry of activity in the use of ML for nowcasting (e.g. Ravuri et al., 2021), however, since the focus
of this review is on weather and climate applications, these studies will not be elaborated on.
**5.1. Early work – 1D deterministic models**
Work on the use of ML to predict chaotic time-domain systems initially focused on 1-D problems, including 1-D
Lorenz systems (e.g. Karunasinghe & Liong, 2006; Vlachas et al., 2018). Of particular interest is Vlachas et al. (2018),
who used Long Short-Term Memory Networks (LSTMs[†]), which are well-suited to complex time domain problems.
Convolutional LSTMs (ConvLSTMs), which combine convolutional layers with an LSTM mechanism, were
introduced in the meteorological domain by Shi et al. (2015) for precipitation nowcasting. They have since seen wide
adoption in other areas (e.g., Yuan et al., 2018; Moishin et al., 2021; Kelotra & Pandey, 2020). Their success in other
domains suggests that revisiting their utility for weather and climate modelling could be worthwhile.
**5.2. Moving to spatially extended deterministic ML-based models**
Replacing a GCM entirely with an ML alternative was first suggested and tested in a spatially-resolved global
configuration by Dueben and Bauer (2018), although for this study they only sought to predict a single variable
(geopotential height at 500 hPa) on a 6 degree grid. Scher (2018) trained a CNN to predict the next model state of a
GCM based on the complete state of the model at the previous step (i.e., an emulator of the GCM). Since this work
was intended to be a proof-of-concept, the authors used a highly simplified GCM with no seasonal or diurnal cycle,
no ocean, no orography, a resolution of ~625 km in the horizontal, and 10 vertical levels. Nonetheless, their ML model
showed impressive capabilities; it was able to predict the complete model state several timesteps ahead, and when run
in an iterative way (i.e., by feeding the model outputs back as new inputs) was able to produce a stable climate run
with the same climate statistics as the GCM, with no long-term drift (even though no conservation properties were
explicitly built into the CNN). Scher & Messori (2019) then extended on this, but continued the proof-of-concept
approach. They investigated the ability of NNs to make skillful forecasts iteratively a day at a time to a lead time of a
few days for GCMs of varying complexity, and explored a combination of other factors, including number of training
years, the effects of model retuning, and the impact of a seasonal cycle on NN model accuracy and stability.

Weyn et al. (2019) aimed to predict a limited number of variables, focusing on the NWP to medium range time domain. They trained a CNN to predict 500 hPa geopotential height and 300 to 700 hPa geopotential thickness over the Northern Hemisphere to up to 14-days lead time, showing better skill out to 3 days than persistence, climatology, and a dynamics-based barotropic vorticity model, but not better than an operational full-physics weather prediction model. Weyn et al. (2020) then improved on this significantly, with a Deep U-Net style CNN trained to predict four variables (geopotential height at 500 and 1000 hPa, 300 to 700 hPa geopotential thickness, and 2 m temperature) globally to 14 days lead time. A major innovation in this study was their use of a cubed-sphere grid, which minimized distortions for planar convolution algorithms while also providing closed boundary conditions for the edges of the cube faces. Additionally, they extended their previous work to include sequence prediction techniques, making skillful predictions possible to longer lead times. Their improved model outperformed persistence and a coarse resolution comparator (a T42 spectral resolution version of the ECMWF IFS model, with 62 vertical levels and ~2.8 degree horizontal resolution) to the full 14 days lead time, but was not as skillful as a higher resolution comparator (a T63 spectral resolution version of the IFS model with 137 vertical levels and ~1.9 degree horizontal resolution) or the operational subseasonal-to-seasonal (S2S) version of the ECMWF IFS.

Clare et al. (2021) tackled a short falling of many of the ML weather and climate models developed to this point, namely that most were deterministic, limiting their potential utility. To address this, they trained a NN to predict full probability density functions of geopotential height at 500 hPa and temperature at 850 hPa at 3 and 5 days lead time, producing a probabilistic forecast which was comparable in accuracy to Weyn et al. (2020).

Choosing to focus on improved skill rather than the question of probabilistic vs deterministic models, Rasp & Thuerey (2021) developed a ResNet model trained to predict geopotential height, temperature and precipitation to 5 days lead time and assessed it against the same set of physical models as Weyn et al. (2020). Their model was close to as skillful as the T63 spectral resolution version of the IFS model, and had better skill to the 5 day lead time than Weyn et al. (2020).

Keisler (2022) took an ambitious step forward, training a Graph Neural Network[†] (GNN) model to predict 6 physical variables on 13 atmospheric levels on a 1-degree horizontal grid, which the authors claim is ~50-2000 times larger than the number of physical quantities predicted by the models in Rasp & Thuerey (2021) and Weyn et al. (2020). Their model worked by iteratively predicting the state of the 6 variables 6 hours into the future (i.e., the output of each model timestep was the input into the next timestep), to a total lead time of 6 days. The authors showed that their model outperformed both Rasp & Thuerey (2021) and Weyn et al. (2020) in the variables common to all three studies. They suggested that the gain in skill seen over previous studies was due to the use of more channels[†] of information, and the higher spatial and temporal resolution of their model. Finally, they showed that their model was more skillful than NOAA's GFS physical model to 6 days lead time, but not as skillful as ECMWF's IFS.

Lam et al. (2022) also used GNNs to build their ML-based weather and climate model, GraphCast. This model was the most skillful ML-based weather and climate model at the time of writing this review. While the first ML-based weather and climate model to claim to exceed the skill of a numerical model was Pangu-Weather (Bi et al., 2022; described in greater detail in the following subsection), GraphCast exceeded the skill of both the ECMWF deterministic operational forecasting system, HRES, and also Pangu-Weather. Furthermore, Lam et al. (2022) paid

particular attention to evaluating their model and HRES against appropriate measures, and included existing model
assessment scorecards from ECMWF to evaluate them. GraphCast capitalized on the ability of GNNs to model
arbitrary sparse interactions by adopting a high-resolution multi-scale mesh representation of the input and output
parameters. It was trained on the ECMWF ERA5 reanalysis archive to produce predictions of five surface variables
and six atmospheric variables, each at 37 vertical pressure levels, on a 0.25° grid. It made predictions on a 6-hourly
timestep and was run autoregressively to produce predictions to a 10-day lead time. The authors demonstrated that
GraphCast was more accurate than HRES on 90.0% of the 2760 variable and lead time combinations they evaluated.
**5.3. Ensemble generation with ML-based models**
A common criticism of ML approaches to weather and climate prediction is the difficulty of representing uncertainty,
and/or the tails of the distribution of predicted parameters. One common method to represent the range of possible
outcomes (including extremes) under different sources of uncertainty is through a well-calibrated ensemble of
predictions. There are a growing number of examples where ensemble generation is  considered, many of which fall
into the category of full-model replacement.
Weyn et al. (2021) explored probabilistic ML predictions using an ensemble of NNs similar to the single-member NN
described in Weyn et al. (2020). The authors expanded the number of variables predicted from 4 to 6, and produced
forecasts to 6 weeks lead time - considerably longer than any comparable work at the time of writing this review.They
considered a variety of initial condition perturbation strategies, and explored the impact of model error by varying the
initial values of the model weights during training to create a multi-model ensemble. They used a combination of the
multi-model ensemble generation approach and initial condition perturbations to generate a 'grand ensemble' of 320
members. They used established metrics for ensemble performance such as RMSE-spread plots, and found that the
320-member grand ensemble combining the multi-model ensemble with initial condition perturbations performed only
slightly better than the multi-model ensemble alone at 14 day lead times. The skill of the ensemble mean of the system,
a control member, and the full ensemble were assessed against the same metrics from the ECMWF sub-seasonal to
seasonal (S2S) prediction system. Their grand ensemble had lower skill than the S2S system at shorter lead times, but
was comparable in skill at longer lead times. Their skill assessment used standard probabilistic skill measures such as
continuous ranked probability score and the ranked probability skill score, which are not present in the other studies
discussed in this Section.The next major ML model to be tested in an ensemble mode was FourCastNet, presented by
Pathak et al. (2022), who leveraged the work on DeepONets described in Section 4. In particular, the authors used a
type of DeepONet called a Fourier Neural Operator (FNO). FourCastNet produced predictions of 20 variables
(including challenging-to-predict variables such as surface winds and precipitation) on five vertical levels with 0.25
degree horizontal resolution, and had competitive skill against the ECMWF IFS to 1 week lead time. The high
horizontal resolution of their model enabled it to resolve extreme events such as tropical cyclones and atmospheric
rivers, and the speed of the model facilitated the generation of large ensembles (up to 1,000's of members).
The authors explored the potential of their ensemble forecasts by generating a 100-member ensemble from initial
conditions perturbed with Gaussian random noise. They showed that the FourCastNet ensemble mean had lower
RMSE and a higher anomaly correlation coefficient than a single-value prediction at longer lead times (beyond ~3-4
days), although the ensemble mean performed slightly worse than the single value forecast at shorter lead times. The
authors attributed this relative decrease in performance at shorter lead times to the ensemble mean smoothing out fine-
scale features. Unfortunately, the authors did not examine the spread of the ensemble with lead time or evaluate the
model using probabilistic skill metrics (in contrast to Weyn et al., 2021), and while they did consider the capacity of
FourCastNet to predict extremes, they did not do so in an ensemble context.
Hu et al. (2023) improved on the relatively simple ensemble perturbation approach employed by Pathak et al. (2022)
in their model, a Swin (sliding window) Transformer-based Variational Recurrent Neural Network (SwinVRNN).
This model combined a Swin Transformer Recurrent Neural Network (SwinRNN) predictor with a Variational Auto-
Encoder perturbation module. The perturbation module learned the multivariate Gaussian distributions of a time-
variant stochastic latent variable from the training data. The SwinRNN predictor was deterministic, but could be used
to generate ensemble predictions by perturbing model features using noise sampled from the distribution learned by
the perturbation module. Unlike the approach used by Pathak et al. (2022), this strategy ensured that the perturbations
applied at each spatial location in ensemble generation were appropriate for the location and variable in question.
Furthermore, the training strategy employed by Hu et al. (2023) accounted for both the error in the deterministic
predictions and the error in the learned perturbation distribution, effectively optimizing forecast accuracy and
ensemble spread at the same time. The authors assessed both the ensemble spread, and ensemble mean accuracy of
their model, and found that it had a better ensemble spread than simpler alternative ensemble generation strategies.
They also found that it had lower latttude-weighted RMSE than the ECMWF IFS to 5 days lead time for 2m
temperatures and total precipitation. ECMWF data beyond 5 days was not shown, but the SwinVRNN models had
latitude-weighted RMSE values lower than a weekly climatology baseline for three of the four variables shown to 14
days lead time. Bi et al. (2022) achieved a significant milestone with their model Pangu-Weather, the first ML-based
model to perform better than the ECMWF IFS to a lead time of 7 days based on RMSE and Anomaly Correlation
Coefficient (ACC) across several variables including geopotential height and temperature at 500 hPa. While they did
explore the utility of Pangu-Weather for ensemble generation, their approach was more simplistic than that
demonstrated by Hu et al. (2023). Pangu-Weather featured two major innovations over previos contributions to this
space:
1. It used 3D (latitude, longitude and height) input grids trained against 3D output grids. This enabled different
levels of the atmosphere to share information, which was not possible in FourCastNet in spite of predicting
variables on multiple atmospheric levels, because the levels were treated independently. In contrast, Pangu-
weather adopted a 3D convolutional method that the authors name the 3D Earth-specific transformer
(3DEST), which enabled the flow of information both horizontally and vertically.

2. It was made up of a series of models trained with different prediction time gaps. The motivation for this was
that, as noted by the authors, when the goal is to produce forecasts to 5 days (for example), but the timestep
of the basic forecast model is relatively short (e.g. 6 hours), many iterative executions of the model are

required, with the errors of each iteration feeding onto the next. A shorter model timestep results in greater
overall errors (due to more iterations being required to reach the final forecast lead time), and a longer model
timestep reduces this error. Motivated by this, the authors trained several versions of their model to predict
to different timesteps on a single iteration. The overall forecast to a given lead time was then constructed
using the longest possible timesteps. For example, for a 7-day forecast, a 24-hour forecast is iterated 7 times,
whereas for a 23-hour forecast, a 6-hour forecast is iterated 3 times, followed by a 3-hour forecast 1 time,
and 1-hour forecast 2 times. The authors noted that this strategy was not effective to multiweek or longer
timescales; they reported that training the model with a 28-day timestep was difficult, for example, and
suggested that more powerful or complex ML methods would be required to achieve this.
As well as the relatively broad measures of RMSE and ACC, the authors assessed the ability of their system to
represent the intensity and track of selected tropical cyclones. They found that Pangu-Weather predicted the tracks of
the cyclones considered with a high degree of accuracy compared to the ECMWF IFS, however it underestimated
cyclone intensity. The authors attributed this to the training data they used (ERA5) also underestimating cyclone
intensity. As noted above, the authors also explored the potential for producing useful ensemble forecasts. To assess
ensemble predictions, they perturbed the initial state of the system with Perlin noise vectors to produce a 100-member
ensemble of forecasts and calculated the RMSE and ACC of the ensemble mean for selected variables. As in Weyn et
al. (2021), the authors noted that the ensemble mean forecasts performed worse than a single deterministic forecast
for shorter lead times (e.g., 1 day), but better for longer lead times. Unfortunately, as with Pathak et al. (2022), Bi et
al. (2022) did not investigate the properties of the spread of the ensemble or assess its skill using standard probabilistic
skill metrics, and their approach to ensemble generation was much simpler than that of Hu et al. (2023).
As already mentioned above, the skill of Pangu-Weather was exceeded by GraphCast, although Lam et al. (2022) only
assessed GraphCast in a deterministic setting. Nonetheless, there is nothing stopping GraphCast from being used to
generate emsemble forecasts in a manner similar to Pangu-Weather. The authors of this review look forward to a more
in-depth intercomparison of the pure ML models in the literature, including an assessment of their performance for
ensemble predictions.
Although the ensemble systems presented in Weyn et al. (2021) and Hu et al. (2023) had lower overall accuracy than
the other models discussed in this section, they still represented the most comprehensive analysis of the behavior and
performance of ensemble ML models (in terms of considering optimal ensemble perturbation strategies, and
quantifying the ensemble behavior) at the time of writing this review. Further investigation into the best methods to
generate and evaluate pure ML model ensembles would be a highly beneficial contribution to the field.
**5.4. Moving to more extensible models**
As the effectiveness of ML approaches are increasingly demonstrated in the literature, additional factors become clear
in considering these models for both research and application. In a research setting, the ability to readily perform
transfer learning to new problems and reduce training costs will be significant in supporting adoption by other
researchers.
This need for greater flexibility in both the input data sources and predictive outputs of ML weather and climate
models was recognized by Nguyen et al. (2023), who developed a transformer architecture-based ML model called
ClimaX. This model was designed as a foundational model, trained initially on datasets derived from the CMIP6
(Eyring et al., 2016) dataset, and able to be readily retrained to specific tasks using transfer learning. The authors
demonstrated the skill of ClimaX against simpler ML models, and in some cases a numerical model (ECMWF IFS),
for a variety of tasks including weather prediction, sub-seasonal prediction, climate scenario prediction, and climate
downscaling. The authors showed that ClimaX was able to make skillful predictions in scenarios unseen during the
initial CMIP6 training phase. Furthermore, ClimaX used novel encoding and aggregation blocks in its architecture to
enable greater flexibility in the types of variables used for training, and to reduce training costs when a large number of
different input variables were used.

## 5.5. Benchmark datasets for ML weather models

Providing open benchmark data for machine learning challenges has been as transformational for the machine learning
field as improved algorithms, the publication of papers, or improvements in hardware.
As the interest and activity in the use of ML as a potential alternative to knowledge-based numerical GCMs has grown,
the need for consistent benchmarks for the intercomparison of ML-based models has become increasingly clear. Rasp
et al. (2020) addressed this need with the introduction of WeatherBench. On this platform, the authors provided data
derived from the ERA5 archive that has been simplified and streamlined for common ML use cases and use by a broad
audience. They also proposed a set of evaluation metrics which facilitate direct comparison between different ML
approaches, and provided baseline scores in these metrics for simple techniques such as linear regression, some deep
learning models and some GCMs. Since the publication of WeatherBench, more benchmark datasets tailored to other
domains have been created, including RainBench (de Witt et al., 2020), WeatherBench Probability (Garg et al., 2022),
and ClimateBench (Watson-Parris et al., 2022). Weyn et al. (2020) chose datasets and assessment metrics consistent
with WeatherBench to facilitate intercomparison of results. Rasp & Thuerey (2021) directly used the benchmarks
provided by WeatherBench in their assessment. They demonstrated that their model outperformed previous
submissions to WeatherBench, highlighting its value as a tool to allow intercomparability of ML-based weather
models. Other examples of studies using WeatherBench data and analysis methods are Clare et al. (2021) and Weyn
et al. (2021). The parameters of a good benchmark dataset were further elucidated by Dueben et al. (2022), who
provided an overview of the current status of benchmark datasets for ML in weather and climate in use in the research
community and provided a set of guidelines for how researchers could build their own benchmark datasets.
At the time of writing this review, assessments of ML-based models had chiefly (but not exclusively) focused on
simple statistics like globally-averaged RMSE, and not reported in detail on the degree to which they accurately
captured specific processes such as cyclone formation, climate drivers such as the El Nino Southern Oscillation, or
large scale structures such as the jetstreams. A useful contribution from the scientific community would be to better
quantify and articulate a suite of tests and statistics that could form a 'report card' to provide better insight into the
value of new ML models.
It should also be noted that all of the major milestones and high-profile ML models described in this section so far
have relied to some degree or another on reanalysis datasets produced by physics-based models. The provision of
higher resolution and higher quality open datasets have the potential to drive progress in this area as much as, if not
more than, improvements and further research into ML algorithms.

**5.6. A hybrid approach**

Arcomano et al. (2022) present an approach which straddles the theme of this section and that of the following section
(physics-constrained ML models). Following Wikner et al. (2020), they used a numerical atmospheric GCM and a
computationally-efficient ML method called reservoir computing in a hybrid configuration called Combined Hybrid-
Parallel Prediction (CHyPP). Their hybrid model is more accurate than the GCM alone for most state variables to a
lead time of 7-8 days. They also demonstrate the utility of their hybrid model for climate predictions with a 10-year
long climate simulation, for which they showed that the hybrid model had smaller systematic errors and more realistic
variability than the GCM alone.

**5.7. ML for predicting ocean variables**

More recently, greater attention has been paid to the application of ML to the ocean, particularly for seasonal to multi-
year prediction. Initial work in this space focused on directly predicting key indices such as the NINO 3.4 index. For
example, Ham et al. (2019) trained a CNN to produce skillful El Niño Southern Oscillation (ENSO) forecasts with a
lead time of up to one and a half years. A limiting factor for the application of ML to ocean variables is the lack of
availability of observational data for training. To overcome this, the authors used transfer learning[†] to train their model
first on historical simulations, and then on a reanalysis from 1871 to 1973. Data from 1984 to 2017 was reserved for
validation. Ham et al. (2021) improved on this by including information about the current season in the network inputs
as one-hot vectors[†]. Including this seasonality information led to an overall increase in skill relative to the model in
Ham et al. (2019), in particular for forecasts initiated in boreal spring, a season which is particularly difficult to predict
beyond.
Kim et al. (2022) improved on the performance of the 2D CNNs used in Ham et al. (2019) and Ham et al. (2021) for
predicting ENSO by instead using a convolutional LSTM network with a global receptive field [†]. The move to a larger
(global) receptive field for the convolutional layers enabled the network to learn the large-scale drivers and precursors
of ENSO variability, and the use of a recurrent[†] architecture (in this case LSTM) facilitated the encoding of long-term
sequential features with visual attention[†]. This led to a 5.8% improvement of the correlation coefficient for Nino3.4
index prediction and 13% improvement in corresponding temporal classification with a 12-month lead time compared
to a 2D CNN.
Taylor & Feng (2022) moved from prediction of indices to spatial outputs, training a Unet-LSTM[†] model on ECMWF
ERA5 monthly mean Sea Surface Temperature (SST) and 2-m air temperature data from 1950-2021 to predict global
2D SSTs up to a 24-month lead time. The authors found that their model was skillful in predicting the 2019-2020 El
Niño and the 2016-2017 and 2017-2018 La Niñas, but not for the 2015-2016 extreme El Niño. Since they did not
include any subsurface information in their training data (in contrast to Ham et al. (2019) and Ham et al. (2021), who
included ocean heat content), they concluded that subsurface information may have been relevant for the evolution of
that event.
It is clear from the small number of (but rapidly evolving) studies in this space that there is great promise for the use
of ML for seasonal and multi-year prediction of ocean variables, with many avenues to pursue to achieve potential
skill gains.
**5.8. ML for climate prediction**
The literature on the use of ML for prediction on seasonal to climate timescales is still relatively sparse compared to
its use for nowcasting and weather prediction. Some examples have been covered in previous sections, such as Weyn
et al. (2021) on subseasonal to seasonal timescales in the atmosphere, and Ham et al. (2019), Ham et al. (2021), Kim
et al. (2022) and Taylor & Feng (2022) on seasonal to multiyear timescales in the ocean. A major cause for this sparsity
is that deep learning typically requires large training datasets, and the available observation period for the earth system
is too short to provide appropriate training data for seasonal to climate timescales in most applications. On the
subseasonal to seasonal end, this may be overcome by including more slowly-varying fields in the training (e.g. ocean
variables), by designing models to learn the underlying dynamics which drive long-term variability, and by including
more physical constraints on the models. On the climate end these same methods could be beneficial, as well as
transfer learning, as is done in Ham et al. (2019), and data augmentation[†] techniques. Additionally, interest is
increasing in the use of ML to predict weather regimes and large-scale circulation patterns, which may prove beneficial
in informing seasonal and climate predictions (Nielsen et al., 2022). Watson-Parris (2021) argued that the differences
between NWP to multiyear prediction and climate modelling mean that the ML approaches best suited to each can be
very different. This may also help to explain why the rapid pace of advances in ML based weather models has not
translated into a similar trend in climate modelling.
Despite this, with the growing maturity of the field of ML for weather and climate prediction, there is every reason to
believe the challenges of prediction on seasonal to climate timescales can be overcome.
**6. Physics constrained ML models**
As has been briefly touched on in previous sections, a promising and increasingly popular method for improving the
performance of ML applications in weather and climate modelling is to include physics-based constraints in the ML
model design (e.g. Karpatne et al., 2017; de Bézenac et al., 2017; Beucler et al., 2019; Yuval et al., 2021; Beucler et
al., 2021; Harder et al., 2022). This can be done through the overall design and formulation of the model, and through
the use of custom loss functions which impose physically-motivated conservations and constraints.
An excellent review of the possible methods for incorporating physics constraints into ML models for weather and
climate modelling, along with 10 case studies of noteworthy applications of these methods, is presented in Kashinath
et al. (2021). The scope of Kashinath et al. (2021) is broad and includes studies not applied directly in the context of
weather and climate modelling, but applicable to it. Rather than repeat the total of this summary here, the reader is
directed to this review.
A class of physics-leveraged ML which has grown rapidly in popularity is Physics Informed Neural Networks
(PINNs). These are discussed in Kashinath et al. (2021), but have also become a very active area of research since the
publication of that review. A more up-to-date review of this class of NNs is presented by Cuomo et al. (2022), along
with a review of other related Physics guided ML architectures.
While PINNs are an exciting and promising new NN architecture, they still face some challenges. For example, they
have had little success simulating dynamical systems whose solution exhibits multi-scale, chaotic or turbulent
behavior. Wang et al. (2022b) attributed this to the inability of PINNs to represent physical causality, and developed
a solution by re-formulating the loss function of a PINN to explicitly account for physical causality during model
training. They demonstrated that this modified PINN was able to successfully simulate chaotic systems such as a
Lorenz system, and the Navier-Stokes equations in the turbulent regime; something which traditional PINNs were
unable to do.
Nonetheless, recent work with PINNs has led to some interesting results for weather and climate simulation: Bihlo &
Popovych (2022) used PINNs to solve the shallow-water equations on a rotating sphere, as a demonstration of their
utility in a meteorological context, and Fuhg et al. (2022) developed a modified PINN to solve interval and fuzzy
partial differential equations, enabling the solving of PDEs including uncertain parameter fields.
**7. Other applications of ML and considerations for the use of ML in Weather and Climate Models**
Aside from the most active areas of development in the use of ML in weather and climate models discussed in the
sections above, there are a few areas of the literature worth mentioning that are adjacent to the main focus of this
review. These topics are covered in the following subsections.
**7.1. Nudging**
Rather than replacing a component or components of a GCM with an ML alternative to gain skill improvements, Watt-
Meyer et al. (2021) focused on using corrective nudging to reduce model biases and the errors they can introduce
through feedbacks. The authors used RFs to learn bias-correcting tendencies from a hindcast nudged towards
observations. They then coupled this RF to a prognostic simulation and attempted to correct the model drift with the
learned nudging tendencies. While this simulation ran stably over the year-long test period and showed improvements
in some variables, the errors in others were observed to increase. So far studies in this space seem to be limited to
Watt-Meyer et al. (2021), however this method seems promising, so hopefully interest in developing this approach
further will grow in the future.
**7.2. Uncertainty quantification**
A common criticism of some ML models such as NNs is that it is difficult to represent the uncertainty of their outputs.
Some examples of studies that have sought to overcome this have already been mentioned in Section 3.8, and there

are other examples in the literature (e.g. Grigo & Koutsourelakis, 2019; Atkinson, 2020; Yeo et al., 2021; O'Leary et al., 2022), however it is nonetheless still a relatively underexplored aspect of ML models for physical systems. Psaros et al. (2022) suggest that this may be because they are also under-utilized within the broader deep learning community, and it is thus a developing field that is not universally trusted and understood yet. They also point out that the physical considerations inherent to ML applied to physical systems often make them more complicated and computationally expensive than standard ML applications, further disincentivizing the inclusion of uncertainty quantification in an already complex problem.

Only recently has attention to this aspect of ML become sufficient to motivate the collection of methods into a consistent framework, a good example of which is the aforementioned Psaros et al. (2022), who presented a comprehensive review of the methods for quantifying uncertainty in NNs and provided a framework for applying these methods.

A related topic which is facing similar challenges is the question of explainability of ML approaches; often there is value in understanding the relative roles and importance of predictors in an ML model, or the relative significance of different regions of the predictor data. Flora et al. (2022) provide a good overview of approaches to this and compare their relative drawbacks and benefits.

**7.3. Capturing extremes**

While there is now an abundance of examples of ML being used for model parameterization schemes, full model replacement, downscaling, and PDE solvers (much of which is covered in this review), there are relatively few examples which address the question of how well ML approaches can reproduce extreme events and statistics, both in terms of the distribution of values predicted in a single-member (i.e., non-ensemble and non-probabilistic) ML model and in terms of the distribution of predicted outcomes in a probabilistic or ensemble ML model.

Both Pathak et al. (2022) and Bi et al. (2022), introduced in Section 5.2, investigated the ability of their models to correctly represent extremes, using a similar approach. They divided their test dataset into 50 percentile bins (distributed logarithmically by Pathak et al. (2022) and linearly by Bi et al. (2022)) between the $90^{th}$ and $99.99^{th}$ percentiles, and computed the relative quantile error between their forecast and ground-truth as a function of lead-time. Pathak et al. (2022) note that they set their highest percentile bin at 99.99% because of the small sample of datapoints beyond this percentile making a statistically significant analysis difficult. Both Pathak et al. (2022) and Bi et al. (2022) found that their models consistently under-forecast extremes to a greater degree than the ECMWF IFS.

Watson (2022) presents a strong argument for the need for a greater focus on the ability of ML weather and climate models to be able to predict extremes in order for them to meet the needs of users. They present a summary of some examples of ML models which have sought to predict extreme events according to certain return period definitions. The example most relevant for this review is Lopez-Gomez et al. (2023), who used a NN with a custom loss function that preferentially weighted extremes to predict global extreme heat. They found that their custom loss function led to improved representation of the tails of the distribution (i.e., predictions of extreme heat), and, interestingly, did not result in any major loss of performance for the middle of the distribution.

The under-prediction of extremes seen in Pathak et al. (2022) and Bi et al. (2022) is consistent with the findings of
Lopez-Gomez et al. (2023), given that neither were not optimized for predicting extremes. These findings all point to
the idea that in order for ML weather and climate models to be able to skillfully predict extreme events, model training
regimes, loss functions and architectures will need to be employed which take into consideration ways to optimize for
these regimes.
**7.4. Object identification within models**
An alternative to achieving greater model accuracy and skill for predicting extremes through increasing resolution of
the entire model grid is to develop techniques to identify critical systems and physical phenomena within the model,
and embed higher resolution temporary subgrids or specialized models within the larger GCM to more accurately
simulate those processes. A challenge to overcome to achieve this is automatically identifying key model features,
since it typically requires a labelled dataset. This requirement can however be avoided, and a variety of both supervised
and unsupervised machine learning approaches to object detection have been demonstrated in the literature.
Mudigonda et al. (2017) were a relatively early example of the application of ML to this challenge. They investigated
the feasibility of using a variety of NN architectures to identify storms, tropical cyclones and atmospheric rivers within
model data, with promising results. Prabhat et al. (2021) provided a valuable resource to the community with their
development of ClimateNet, a labelled open dataset and ML model for the segmentation and identification of tropical
cyclones and atmospheric rivers. This was used by Kapp-Schwoerer et al. (2020) to train a NN to identify and track
these extreme events in Community Atmosphere Model 5 (CAM5; Conley et al. 2012) data. O'Brien et al. (2021)
considered the need for uncertainty quantification in object identification, using a Baysean approach to build an
atmospheric river detection framework. Finally, Rupe et al. (2023) took a physics-informed approach to object
detection, defining 'local causal states' using speed-of-light causality arguments to identify regions of organized
coherent flow and bypassing the requirement for labelled datasets. They demonstrated the utility of their approach for
the unsupervised identification and tracking of hurricanes and other examples of extreme weather events.
While there are unsupervised learning approaches which have shown value for object detection in weather and climate
data (e.g. Rupe et al., 2023), a major limitation of this area of research is the shortage of labelled datasets for supervised
learning methods , with ClimateNet being an isolated example.
**7.5. GPUs and specialized compute resources**
GPUs and TPUs are specialized hardware which are well suited to highly parallelizable matrix operations, ideal for
solving neural network operations. TPUs have been developed specifically for deep learning applications. Both GPUs
and TPUs are likely to be available on many of the next generation of supercomputers, but much of the current Fortran-
based numerical weather and climate model infrastructure cannot be run on them in their current state. Data
bottlenecks also exist between the GPUs (which have their own on-board memory) and the main memory accessible
to the CPU. While efforts are underway to make numerical and climate models better suited to GPUs, for example
with the development of LFRic (Adams et al. 2019), the new weather and climate modelling system being developed
by the UK Met Office to replace the existing Unified Model (Walters et al. 2017), there is still a long way to go before
entire weather and climate models can be reliably run on GPU or other specialized compute architectures. At the same
time, some neural network designs are aimed squarely at the partial differential equation solving at the core of
numerical methods. Since neural network evaluation utilizes simpler mathematical operations than current PDE
solvers, they offer the prospect of significant computational advantages on non-specialized (i.e., CPU) hardware.
**8. Perspectives on machine learning from computer science**
This section provides a brief perspective on weather and climate modelling from the computer science domain and
aims to provide the earth system scientist with a short list of the main relevant innovations in computer science. As
was noted in Section 1, ML models are often regarded as black-boxes, largely because of the design of many prominent
ML systems. In principle, it is not quite right to refer to the trained model as "a machine learning model", in the sense
that the process of training the model is "machine learning", once the model is trained it is definable by a set of
mathematical equations and coefficients, much like any physical, statistical, or theoretical model. Thus the machine
learning refers to the training process, not the model itself.  The essence of ML is the level of automation involved.
Even in typical ML models such as large NNs, the model architecture is typically specified manually by the data
scientist or physical scientist involved. The automated derivation of model architecture and composition is not yet
mature for large models, although it is explored through evolutionary programming techniques whereby the learning
of architecture as well as parameterization is automated.
The complex nature of the Earth system means that ML models which seek to emulate it (or subcomponents of it) will
likely also need to be quite complex, and will contain a mixture of ML architectures and algorithms. This is borne out
by the increasing degree of complexity and variety seen in the ML models in the literature reviewed in previous
sections.
A large degree of the current research focus is on very large or deep NNs which rely both on the universal
approximation theorem and practical experimentation to capture a prediction function without needing to explicitly
represent the processes being modeled. In a conceptually similar fashion to how a Fourier decomposition can represent
any wavelike function, the universal approximation theorem establishes that a NN may approximate any function,
subject to its size and the required degree of accuracy (Hornik, Stinchcome and White 1989). Deep learning has been
highly effective in approaching many problems, but many limitations are acknowledged, as evidenced by the current
widespread focus on trustworthy computing and efforts towards explainable ML systems. Some ML models take a
direct approach to modelling the uncertainty of the system being simulated by representing the model state variables
as a probability distribution or degree of confidence. Many contemporary weather and climate model derive their
probabilistic outputs from an ensemble of perturbed members, however an alternative approach is to represent each
part of the belief state[†] of the model as a distribution or likelihood, built up either empirically or by fitting a gaussian
or other known distribution (e.g., Clare et al., 2021).
A timeline of some key innovations in ML is presented in Figure 4. The scale of the timeline is broken between 1956

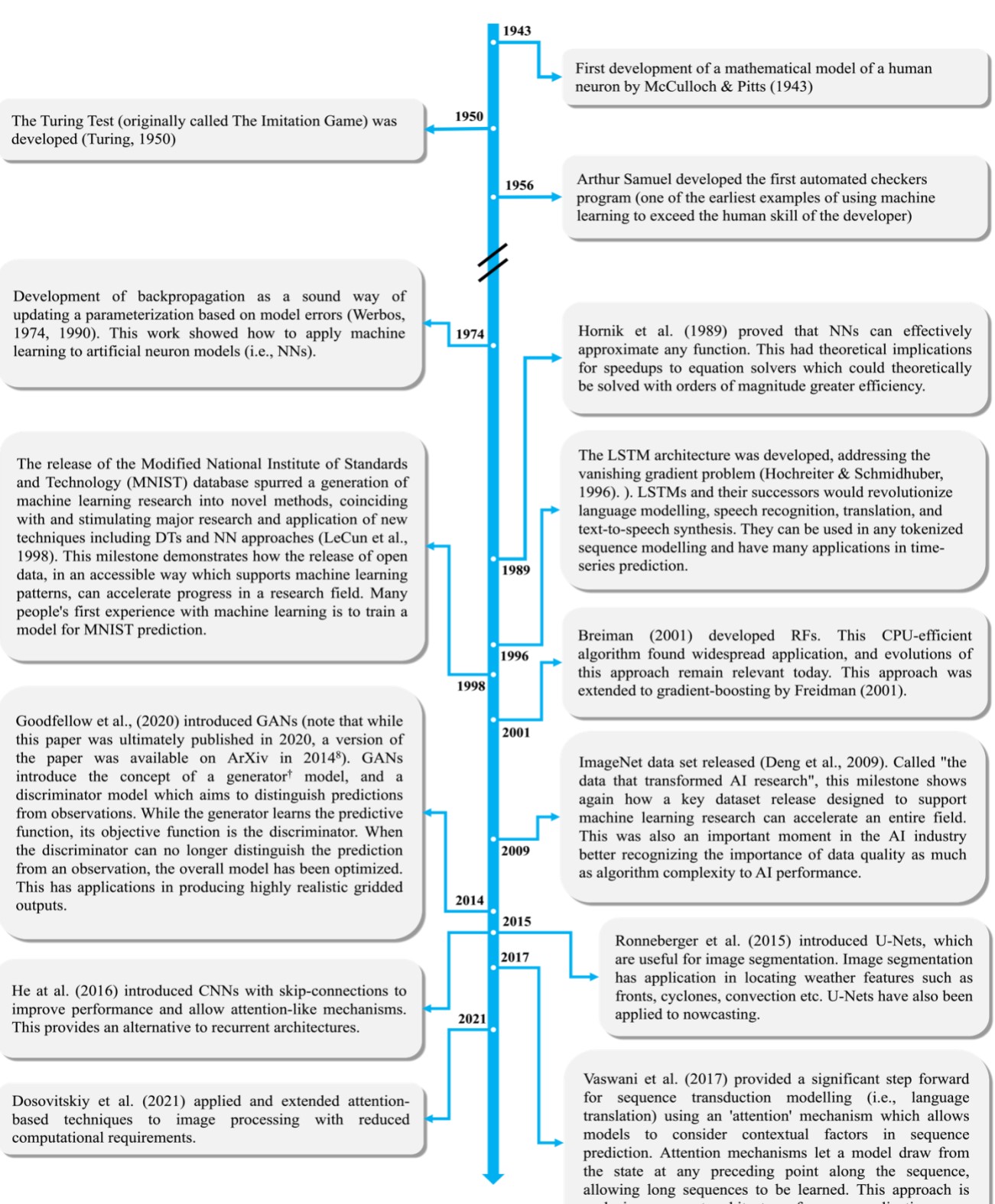

Figure 4: A timeline of key breakthroughs in ML.

and 1974, and Taking that gap in progress into account, it is clear from this visualization that the rate of innovation in
ML has increased significantly over the last 35 or so years. This is likely driven by a range of factors including the
increasing availability of compute resources suited to ML applications, and the explosion of available data for training.
This history shows the degree and rate of research into processing images, text and other sequences based on semantic
understanding of content, but does not demonstrate capturing physical processes as a core element. Advances in the
weather and climate modelling domain have a more explicit goal of properly portraying real physical processes.
Bringing these concepts together promises to uplift capability in both fields.
**9. Practical Perspectives on Machine Learning for Weather and Climate Models**
A major driver of research into, and improvement of, weather and climate models is increasing the skill of operational
forecast systems, and increasing the accuracy and trustworthiness of climate projections. Therefore, an important
consideration for ML in the context of weather and climate models is the need for it to ultimately be integrated into a
complete predictive system with practical application for forecasting or climate projections.
However, the research findings covered in this review, in spite of being compelling, are yet to make major changes to
operational modelling systems, or standard climate projections.
We have identified three major challenges facing the transition of ML-based innovations into operational settings.
Similar challenges are faced in the context of climate projections, however since these are out of scope for this review
we do not discuss them directly, and instead leave them as a topic for other publications.
The first challenge is the need to assess when a research finding is sufficiently compelling and robust to justify
integration into established operational systems. Since the major function of operational meteorological services is to
inform of future conditions, largely for managing risk or optimizing benefits, a conservative approach is taken to
changing these systems. The utmost premium is put on accuracy, resilience, reliability, and solid scientific foundation,
and many novel research finding require extensive further evaluation and development before they can be considered
ready for inclusion into operational systems. Understanding when to invest this degree of effort in bringing a research
innovation into a major model or scientific configuration upgrade can be difficult.
The second major challenge is establishing the right balance between potentially unwieldy monolithic ML models
which predict all variables of interest, and many smaller limited scope models which each focus on predicting one or
a small number of variables well. The former option is more similar to current dynamical systems, while the latter
option is potentially more easily achievable using an ML approach, but risks becoming difficult to manage due to the
proliferation of small, separate systems. The early effectiveness of limited-purpose ML models provides the ability to
augment existing services without disruption, however aside from the logistical complexity of many small systems, a
risk associated with this approach is that inconsistencies between predictions may arise from their independent
forecasts, leading to confusion from users and an erosion of trust.
Finally, the third major challenge is how to best monitor and maintain the skill of ML-based systems in a real-time
operational context. Explainability of ML systems is an emerging field, and is not yet sufficiently mature for
application to real-time operational monitoring. Until this changes, the ongoing trustworthiness of operational ML
systems will be difficult to demonstrate. Similarly, online learning in ML weather and climate models is not yet a well
explored research area. The use of online learning is likely to be important for operational ML models to be able to
develop resiliency and maintain good skill over time, so more work will be needed in this area before these models
can see greater uptake in operational systems.
In addition to these major challenges, agencies looking to incorporate ML components into their operational systems
must consider that:

- the explainability of ML model errors in the case of poor forecasts that may come under scrutiny,

- the robustness of ML models to real-time data issues such as data dropouts or input data degradation must be
established, and

- the lack of infrastructure in these agencies to support ML models in an operational setting will need to be
addressed.
Operational development is typically quite incremental, and it is likely that progress will be made in small achievable
steps along the evolving technical frontier. However promising and fascinating as a research direction, full model
replacement with ML alternatives is currently not mature enough for an operational setting. Instead, the authors predict
that the first types of ML systems to be seen in operations will include parameterization scheme replacements and
emulators, solver replacements, super-resolution, new approaches to data assimilation of novel observation sources,
and both pre- and post-processing applications (although of course not all of these have been covered in this review).
It is expected that the research into, and application of, ML methods will represent a growing proportion of weather
and climate model research, with increasingly sophisticated and skillful model components finding their way into
major model releases over the coming years. These components are appealing for both computational and model skill
reasons, and are expected to be highly promising avenues of research.

## 1150 10. Ethical considerations for Machine Learning for Weather and Climate Models

Not all papers in this review included a discussion of the ethical considerations associated with using machine learning,
nor necessarily touched on what constitutes a sufficiently rigorous verification methodology for machine learning
models. There is a clear relationship between ethical considerations, the explainability of models, and the rigor of
verification applied to ensure that models behave as expected under a variety of conditions (and do not include
unexpected behaviours).
While this review paper does not provide an introduction to AI and ML ethics in general, a brief overview of some
of the important considerations for the application of ML in the context of weather and climate modelling is
provided in this section. Ethical frameworks vary in different cultural and geographical contexts, and for a more
general introduction to the ethical considerations surrounding AI and ML, the reader is directed to the paper
*Recommendations on the Ethics of Artificial Intelligence* (United Nations Educational, Scientific and Cultural
Organisation (UNESCO), 2022).
For ML applied to weather and climate modelling, some considerations to ensure sufficient robustness and reliability
include whether:
• testing, training and validation data sets are sufficiently representative of the data in general
• potential causal correlations between testing, training and validation data have been treated correctly
• trained models have been tested for reliability against adversarial examples
• data augmentation (e.g. noise addition) has been utilized to enhance model robustness
• an evaluation of the potential for model drift has been performed
• the training data is biased in a way which results in ethical unfairness (for example – remote communities
may not receive equal-skill predictions due to a lack of observational training data in remote areas,
• the machine learning method is compared to a suitable alternative, such as a known physical model in
addition to any comparisons to machine learning models or the provision of aggregate statistics
• the data that has been used has been gathered ethically, and any personal information has been treated
properly (such as when processing weather reports from individuals)
• the authors have identified any caveats regarding ethics, reliability, robustness or explainability
• the authors have investigated the physical realism of the predictions from ML models
This list is not comprehensive, however. A thorough overview of the explainability, reliability, ethics, and
verification of ML models in weather and climate has not been covered in prior literature and the field will
benefit from further work in this area.
**11. Future research directions**
The already-demonstrated and potential future applications for ML in weather and climate modelling are significant
in number, and identifying the most fruitful avenues for future research can seem overwhelming. A good
understanding of the current state of the weather and climate modelling field, along with knowledge of the key
developments in ML research, are required to assess the potential benefits of a given research direction.
As can be seen from the timeline of machine learning presented in Figure 4, older techniques can prove to be
relevant many years later, and there are many techniques from computer science which may become relevant for
contemporary weather and climate modelling problems and research.
Furthermore, due to the general applicability of many ML approaches, research progresses in one subdomain may
have implications and benefits for another. For example, DeepONets were developed for, and shown to be
successful for, solving PDEs, but were adopted by Pathak et al. (2022) for their pure ML model FourCastNet with
great success.
To help the reader navigate the myriad research areas where ML for weather and climate modelling could be
progressed, five categories of future research directions are presented in Figure 5, along with some specific areas of
research, and benefits that could arise from them.
These categories are not mutually exclusive – indeed there is overlap between the research areas and benefits
highlighted in each category (for example, some research foci in Categories 2 and 3 are also applicable to Category
5). The groupings are instead intended to help guide the focus of researchers, and to provide a quick overview of the
key topics where the community would most benefit from research progress.


### Category 1: Improving training speed and efficiency

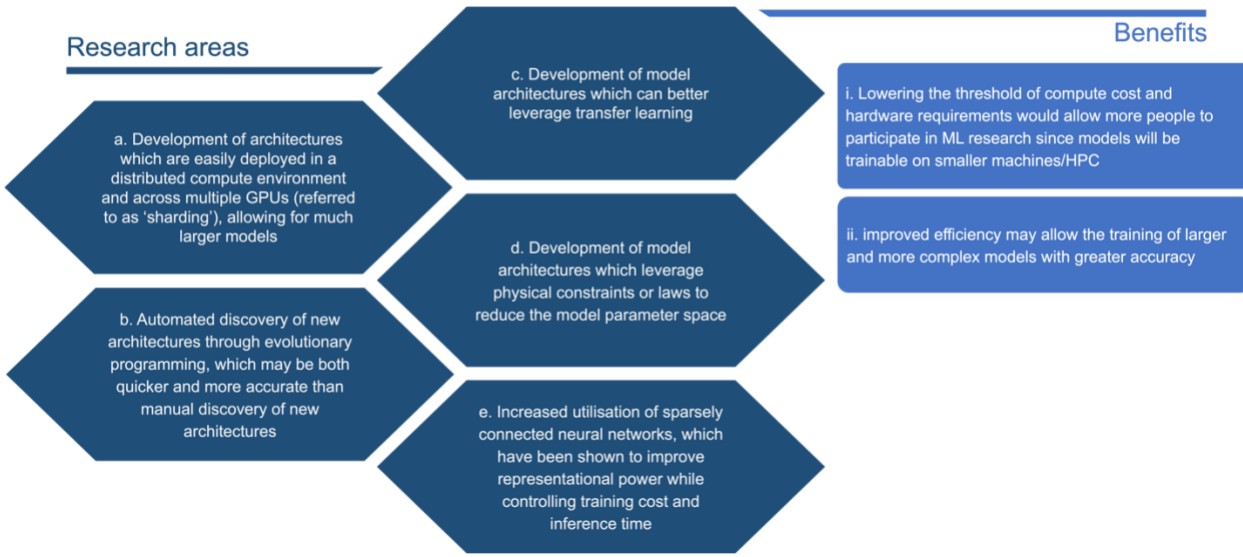


### Category 2: Physically consistent/constrained models and evaluation

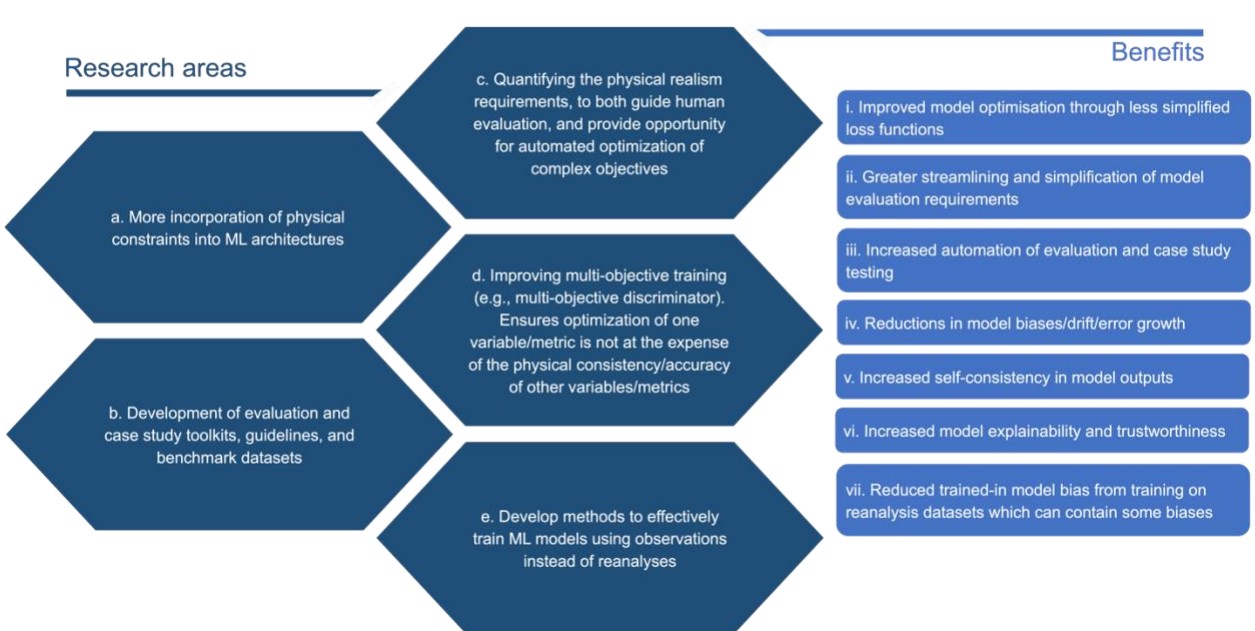


## Category 3: Weather and climate modelling domain specific research

### Research areas

**a.** More ML parametrization schemes for land surface, sea ice, and ocean models. Many process in these models are slow compared to atmospheric processes, so ML parametrizations of them should be more stable

**b.** Expanding the scope of pure ML models to include more variables and components of the earth system

**c.** Development of ML models with multiple components for different physical processes. For example, a ML cloud model could have separate advection and convection components, specialized to predict different dynamical processes independently

**d.** Development of an ML coupler, for example for coupling atmosphere and ocean models, including learning optimal coupling strengths

**e.** Development of an ML component which corrects the model at each integration step to reduce model drift/biases. Superior to postprocessing bias-correction because it reduces errors in feedbacks and teleconnections

**f.** Develop ML architectures which can undertake a range of tasks (e.g. global or regional modelling, weather prediction or climate simulation, etc.), and are stable when performing out of training sample inference

### Benefits

**i.** Improved speed and/or accuracy of parameterization schemes and variables in ocean, sea ice and land surface models

**ii.** ML-based parametrization of fields which were previously externally forced

**iii.** Increased rate of coupled model development due to the reduced time and computational cost of model tuning with an ML-based coupler

**iv.** Reduced model biases leading to improved teleconnections and more accurate predictions

**v.** Skill to longer lead times due to models being trained on a larger sample of physical process timescales (e.g. by including ocean components to improve skill for the atmosphere)

**vi.** Models that work in a variety of scenarios or for a variety of applications, such as climate change, different forcings, different regions or different timescales

**vii.** Increased model explainability and trustworthiness through component-wise simulation of dynamical processes (e.g., separate simulation of advection and convection enabling analysis of the relative contribution to total model error)


## Category 4: Probabilistic prediction

### Research areas

**a.** Exploring methods to improve the spread in ML based ensembles, such as initial condition perturbation, adding stochastic noise, multi-model ensembles, adding samples from learned noise distributions within the model, etc.

**b.** Using ML emulators of existing dynamical models to augment the size of the model ensemble, or to increase the ensemble size of hindcasts/reforecasts

**c.** Using ML based ensemble generation methods to dynamically add ensemble members adjacent to potential extremes, improving probability resolution

**d.** Probabilistic ML parameterization or emulation of subgrid-scale processes to support realistic stochasticity in dynamical models

### Benefits

**i.** The ability to produce very large ensembles cheaply, either through direct ML based prediction, or augmentation with ML based emulators

**ii.** Greater resolution of the probability distribution curve around extreme events

**iii.** Skilful prediction using ML models to longer lead times, with the same ensemble techniques employed in traditional sub-seasonal to seasonal prediction (i.e., accounting for uncertainty with ensemble spread, and averaging out random noise via an ensemble mean)

**iv.** The ability to directly model the pdf of model variables

**v.** Increased reliability of earth system models during extreme events through large ML based ensembles


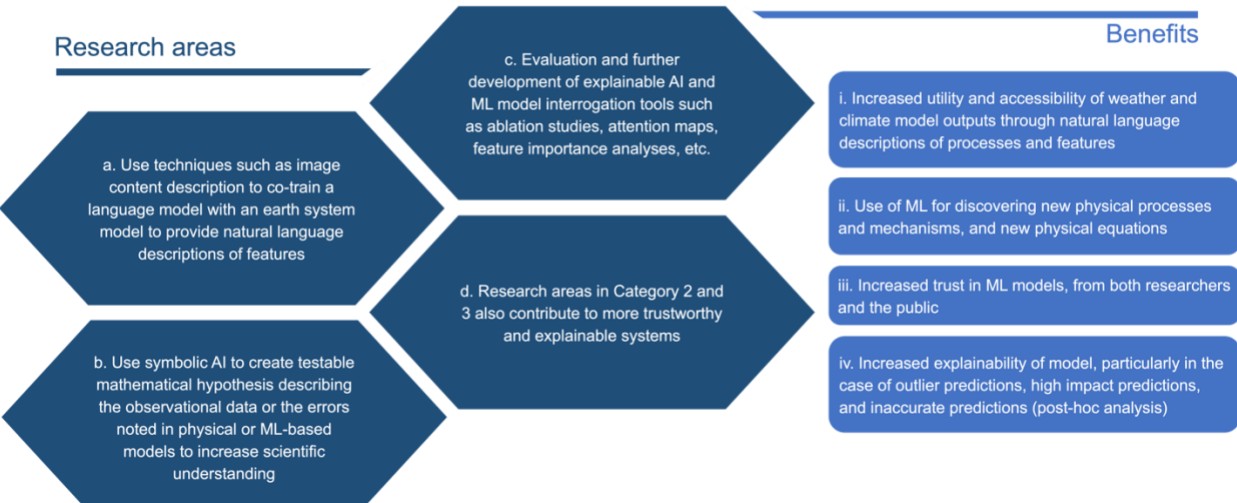

Figure 5: Five categories for future ML research, including suggested research focusses for the community in each category, and potential benefits which could be realized by research and development progress.

Many of the research areas presented are complementary to each other, for example progress in making ML models more affordable to train (Category 1) will increase the utility of ML solutions to a wider community of researchers, and will likely accelerate the rate of progress in the other categories. Progress in the use of physically-informed approaches (e.g. Category2, area a., or Category 3, area c.) could also lower the training cost of models by reducing the degree of redundancy in the model. On the other hand, approaches such as Category 3, area f., leading to an outcome such as benefit vi. would potentially reduce the demand for more cheaply trainable models, since they could be readily turned to a variety of tasks, saving researchers the need to train their own model from scratch. The research areas and ideas presented here are by no means a comprehensive list. Rather they are intended to be used as a source of inspiration, and the authors of this review are excited to see where the community chooses to focus their efforts in the coming years.

## 12. Conclusions

In this review we have presented a comprehensive survey of the literature on the use of ML in weather and climate modelling.

We have found that the ML models being most often explored include RFs and NNs, with a high prevalence of FCNNs and CNNs. We have also identified some recent innovations which have proven to be highly effective in the weather and climate modelling space, including DeepONets and variants thereof, Graph NNs, and PINNs.

This review has demonstrated that ML is being successfully applied to many aspects of weather and climate modelling. We have presented examples from the literature of its application in (1) the emulation and replacement of subgrid-scale parametrizations and super-parametrizations, (2) preconditioning and solving of resolved equations, (3) full model replacement, and (4) a selection of other adjacent areas.

Nonetheless, there are still many research challenges to overcome, including:

- addressing the instabilities excited in physical models due to the inclusion of ML components;
- increasing the ease of technical integration (in particular, Fortran compatibility);
- memory and computational concerns;
- representing a sufficient number of physical parameters and increasing physical and temporal resolution in ML-based weather and climate model implementations (which currently feature reduced fields and levels compared to physics-based numerical models);
- moving from a focus on individual parts of the earth system (i.e., the atmosphere, the ocean, the land surface etc.) to tackling the challenges associated with coupled models (i.e., where models of individual components of the earth system are coupled together). Increasingly, operational weather and climate models are coupled land-atmosphere-ocean-sea-ice models in order to more accurately represent the relevant timescales and processes in the earth system, and ML modelling efforts need to reflect this;
- more thorough evaluation of the physical realism of ML-based predictions, at various length-scales, across parameters, and looking at the three-dimensional structures
- Exploring the use of generalized discriminators to augment traditional loss functions in model training (to achieve a multivariate generalized objective function)
- the need for more good quality training data; and
- the practical challenges of integrating ML components or models into an operational setting.

This list, together with Section 11, provides a set of focus areas for future research efforts.

If the current trend in skill gains in full ML weather and climate models continues, it is possible they will eventually be considered viable alternatives to traditional numerical models. However, in the meantime it is likely that ML components will replace an increasing number of physics-based model components, with models the near-term future being hybrid ML-physical models. A likely future scenario is one where the best weather and climate models are a blend of ML and physics-based components, deriving skill from both data driven and physical methodologies.

Some possible avenues through which increases in ML-based weather and climate model skill might be achieved is by operating at higher resolutions, resolving more processes which are implicit in the training data, or by undertaking experiments on synthetic data to address the paucity of real-world data.

Another benefit of ML approaches to weather and climate modeling is the relative computational cheapness of ML alternatives to current physics-based modelling systems. This has the potential to open the door to experiments that would not be feasible otherwise. For example, experiments requiring a very large ensemble would be more feasible with a computationally cheap ML approach.

The literature reviewed here indicates that 'out of the box' ML approaches and architectures are not effective when used in a weather and climate modelling context. Rather, ML architectures must be adapted to satisfy conservation of energy, represent physically realistic predictions and processes, and maintain good model stability. At the same time, computational and memory tractability must be maintained.

Advances in the sophistication, complexity and efficiency of ML architectures are being heavily invested in for many use cases in other disciplines and in the private sector (e.g., condition-action pose estimation, text to video generation, stable diffusion/text to image, chatbots, facial recognition, semantic image decomposition, etc.). In order to capture the full benefits of ML for the weather and climate modelling domain, academic and operational agencies will need to continue to support research in this space. This includes contributing to the research effort through foci such as those highlighted in Section 11 and in this section, and through addressing the particular challenges facing agencies interested in the operational and/or realtime deployment of ML based models as the basis for services or the provision of advice (discussed in Section 9).

Interest and progress in the application of ML to weather and climate modelling has been present for close to 30 years, and has begun to accelerate rapidly in the last few years. There is good reason to believe that ML as a tool will have transformational benefits  and offers great potential for further application in weather and climate modelling.

**Machine Learning Glossary of Terms**

This glossary includes terms which the reader will come across frequently in machine learning literature for the weather and climate, as well as in machine learning literature generally. Most of these terms are used in this paper while others support further reading.

**Activation Function.** The function which produces a neuron's outputs given its inputs. Commonly, this includes a learned bias term which is added to the data inputs before evaluation with a single function to produce the output value. Examples of the functions used include linear, sigmoid and tanh.

**Adversarial attack.** The deliberate use of malicious data input in a real-world setting intended to cause a misclassification, underperformance or unexpected behaviours. Examples include emails designed to avoid spam filters, or images that have been modified to avoid recognition.

**Adversarial example.** A specialised input which results in a misclassification or underperformance of a predictive model. An example of this concept is an image which has had subtle noise added to it resulting in a copy of that image which is visually indistinguishable from the original, but which nonetheless causes a misclassification. The term 'adversarial' is used to refer to the way the example fools the model and is not necessarily intended to convey the sense of malicious intent, although the term is often applied in that fashion. Adversarial examples demonstrate that machine learning models may be more brittle than expected based on ordinary training data alone. To increase model robustness, adversarial examples may be generated and added to the training set. Data augmentation techniques such as flipping, warping and adding noise (any many other techniques) are also used to generate additional training data to increase robustness and performance.

**Attention mechanism.** A mechanism to allow sequence prediction models to increase the importance of key terms within that sequence which may be nonlocal and modified in meaning according to the other terms of the sequence.

**API**. Application Programming Interface. A set of programming functions, methods or protocols by which to build and integrate applications. APIs may be "web" APIs or imported from software packages in which case they are more often referred to as libraries.

**Autoencoder.** A neural network architecture which learns to produce a 'code' for an input sequence from which the original data can be retrieved. The code is shorter than the original input sequence. Applications include data compression and denoising data.

**Back propagation.** A process of utilising the errors from a prediction to update the weights and biases of a neural network.

**Batch.** See training batch.

**Batch normalisation.** Data normalisation which aligns the means and variances of input data to a model. For computational reasons, this is performed separately for each training batch.

**Belief state.** The current state of the world which is believed to be true according to a model. A common architecture in realtime applications whereby a belief state is updated according to an update function on the basis of new observations.

**Channel.** An additional dimension to data which is usually not a spatial dimension. Examples include the red, green and blue intensity images which comprise a colour image. Another example could be to represent both temperature and wind speed as channels.

**Classification.** A model which attempts to diagnose or predict the category, label, class or type that an example falls within.

**Climatology.** Refers to the usual past conditions for a location at a time of year. Usually calculated by temporal mean across years of a dataset, for a given time interval within those years (e.g., for a dataset of monthly mean values spanning all months of all years from 1990 to 2020, the monthly mean climatology would be obtained by averaging across all the Januarys from each year, all the Februarys, etc., to obtain an "average January", an "average February", etc.). Climatologies are often used in the same manner as persistence as a baseline prediction against which to measure a predictive model. For example, a model predicting a value for January could be compared to the climatological monthly mean value for January. This helps answer the question "is my model a better source of information than using the average past conditions from this time of year?".

**Connectome.** The connections between nodes in a neural network. Examples include fully-connected, partially-connected, skip-layer connections, recurrent connections and others. The 'wiring diagram' for the network.

**Convolutional neural network.** A neural network architecture commonly applied to images which utilises a convolutional (spatially connected) kernel applied in a sliding window fashion with a narrow receptive field to encourage the network to generalise from fine scale structure to higher levels of abstraction.

**Data augmentation.** The practice of modifying input data in supervised learning to produce additional examples. This can make networks more robust to new inputs and address issues of brittleness to adversarial examples. An example of data augmentation is using rotated or reflected versions of the same image as independent training samples.

**Data driven**. A generalised term used to indicate a primary reliance or dependence on the collection or analysis of data. Used in contrast to process driven or theory driven.

**Decision tree.** A tree-like, or flowchart-like, branching model representing a series of decisions and their possible consequences. Each internal node represents a 'test' (i.e. decision threshold) and each leaf node represents a class label or collection of possible outcomes.

**Deep NN.** A neural network with many layers. Deeper, thinner networks have generally been more popular in recent
times than wider, shallower ones but this is not always the case (see e.g. Zagoruyko & Komodakis, 2016)
**DeepONet.** A neural network architecture relying on universal approximation theorem to train a neural network to
represent a mathematical operation (the operator), such as a partial differential equation or dynamic system.
**Discriminator model.** A model which distinguishes or discriminates between synthetic data and real-world
observations. Often used in conjunction with a generator. In this case, the overall goal is to produce a generator which
is capable of fooling the discriminator, producing highly realistic images. This process is used in Generative
Adverserial Networks.
**Dropout layer.** A neural network layer which is only partially connected, often with a stochastic dropout chance. This
has been shown experimentally to improve neural network robustness in many architectures by reducing overfitting.
**Epoch.** A single complete training pass through all available training data, e.g. learning from all samples, or learning
from all mini-batches, according to the training strategy. Multiple training epochs will typically be utilised although
alternative strategies do exist.
**Feed-forward network.** A neural network composed of distinct 'layers', where the outputs of one layer never feed
back into earlier layers. This avoids the needs for any iterative solver approaches and results in a very computationally
efficient 'forward pass'.
**Generative adversarial network.** A two-part neural network architecture comprising a generator and a discriminator,
which are co-trained to produce realistic outputs which are hard to distinguish from real-world data. The discriminator
replaces the traditional loss function.
**Generator model.** A model which produces a synthetic example of a particular class, such as a synthetic image or
synthetic language. Examples include language or image generation. These are used as part of Generative Adverserial
Networks among other applications.
**Global receptive field.** Where every part of the input region can influence or stimulate a response in a model (e.g. a
fully-connected neural network).
**GPU.** Graphical Processing Unit. A hardware device specialised for fast matrix operations, originally created to
support computer graphics, particularly for games.
**Gradient boosted decision tree.** Also referred to as extreme gradient boosting. A random forest architecture which
combines gradient boosting with decision tree ensembles.
**Gradient boosting.** An approach to model training where each additional ensemble member attempts to predict the
cumulative errors of previously trained members.
**Graph neural network.** A class of neural networks designed to process data which is described by a graph (or
tree/network) data structure. See Scarselli et al. (2008), Kipf & Welling (2016), and Battaglia et al. (2018) for more
information and examples.
**Hidden layer.** A layer which is intermediate between the input layer and the output layer of a network or tree structure.
Hidden layers may be used to encode 'hidden variables' which are latent to a problem but not able to be directly
observed.
**Hierarchical temporal aggregation.** A mechanism of composing neural networks which are trained for different lead
times to produce an optimal prediction at all time horizons.
**Hierarchical temporal memory.** Fundamentally different to hierarchical temporal aggregation. A complex deep
learning architecture which uses time-adjacency pooling.
**Hyperparameter.** A parameter which is not derived via training. Examples include the learning rate and the model
topology.
**Hyperparameter search (or Hyperparameter optimization).** The process of determining optimal hyperparameters.
This term may also be used to encompass the model selection problem. This process is automated in some cases.
**Input layer.** A layer which is composed of input nodes. Typically machine learning models will have one input layer
at depth zero (i.e. with no preceding layers) and no input nodes at greater depths.
**Input node.** A node which represents an input or observed value.
**K-fold cross-validation**. A process of changing the validation and test data partitions during different iterations of
training. This allows more of the training and validation data to be used while minimising overfitting. Some definitions
include test data in this process but that is not ideal as the final test is no longer statistically independent.
**Keras.** A streamlined API for creating neural networks, integrated with Tensorflow. Originally built on the Theano
framework for general mathematical evaluation. PyTensor and Aesara are related packages.
**Kernel trick.** For data sets which are not linearly separable, first multiplying the data by a nonlinear function in a
higher dimension can result in a linearly separable higher-dimensional data set to which a simpler method can be used
to model the data.
**Knowledge based systems**. A broad term from artificial intelligence meaning a system which that uses reasoning and
a knowledge base to support decision making. Knowledge is represented explicitly and a reasoning or inference engine
is used to arrive at new knowledge.
**Layer.** In tree or feed-forward network structures (e.g. decision trees and feed-forward neural networks), a layer refers
to the set of nodes at the same depth within a network.
**Leaf node.** Aka output node. A node which does not have any child nodes.
**Long short term memory network.** A recurrent neural network architecture which processes sequences of tokens
utilising a 'memory' component which can store information from tokens early in a sequence for use in prediction of
tokens much later in a sequence. Typical applications include language prediction and time-series prediction of many
kinds.
**Loss function** (also known as target function, training function, objective function, penalty score, error function,
heuristic function, minimisation function). A differentiable function which is well-behaved, such that smaller values
represent better model performance and larger values represent worse performance. An example would be the root-
mean-squared-error of a prediction compared to the truth or target value.
**Mini batch.** A subset or 'mini batch' of the training data. Utilised for multiple reasons, including computational
efficiency and to reduce overfitting. Aggregate error over a mini-batch is be learned rather than per-sample errors.
This is the typical contemporary approach. See also training batch for in-depth discussion.
**Neural network.** A composition of 'input nodes', 'connections', 'nodes', 'layers', 'output layers' and 'activation
functions' which are capable of complex modelling tasks. Originally designed to simulate human neural functioning
and subsequently applied to a range of applications.
**Node. Aka vertex.** A small data structure in a network, tree or graph structure which is connected by edges. A node
may represent a real-world value (such as a location) or an abstract value (such as in a neural network), or a decision
threshold (such as in a decision tree).
**Normalisation.** A technique applied in many areas of mathematics, science and statistics which is also very important
to machine learning and neural networks. In a general sense, this refers to expressing values within a standard range.
Very often, the range of expected values is mapped onto the range 0 to 1, to allow physical variables with different
measurement units to be compared on equal scale. Such normalisation may be linear or nonlinear, according to a
simple or more complex function, and either drawn from known physical limits or from the variation observed in the
data itself.
**One-hot vector.** A vector of 1s and 0s, in which only one bit is set to 1. Typically produced during the first step in
machine learning for language processing to create a word or feature embedding in a process called tokenisation or
encoding. The length of the vector is commonly equal to the number of categories or symbols.
**Output layer.** A layer which comprises the leaf nodes or output nodes of a tree or network.
**Perceptron.** A single-layer neural network architecture for supervised learning of binary classification. Originally
built as an electronic hardware device encoding weights with potentiometers and learning with motors. A multi-layer
perceptron is the same thing as an ordinary neural network.
**Persistence.** Refers to the practice of treating some past observation or reanalysis (usually immediately prior to the
starting point of the prediction period) as the future prediction and "persisting" this one state forward to every
prediction lead time. The predictive model is then compared to this persistence prediction, essentially assessing the
performance of the model against a steady state prediction. This, along with climatology, is often used as a baseline
or bare minimum prediction to beat (i.e., a prediction better than persistence could be considered skilful vs
persistence). This answers the question " is my model a better source of information than using what happened just
before now?".
**Physically-informed machine learning. Also known as physics-informed machine learning.** Machine learning is
considered physically informed when some aspect of physics is included in any way. Examples include adding a
physical component to the loss function (e.g. to enforce conservation of physical properties) or using an activation
function with physically realistic properties.
**Predictive step, forward pass, evaluation.** The process of calculating a model prediction from a set of input
conditions. Distinct from the training phase or back-propagation step.
**PyTorch.** A widely adopted framework for neural networks in Python.
**Random forest.** An architecture based on decision tree ensembles where each decision tree is initialised semi-
randomly and an average of all models is used for prediction. This is typically more accurate than a single decision
tree but less accurate than a gradient-boosted decision tree and so is now less-used. The term random forest is still
commonly used when in fact the implementation is a gradient boosted decision tree.
**Receptive field.** The size or extent of a region in the input which can influence or stimulate a response in a model,
e.g. the size of a convolutional kernel, the size of a sliding window
**Rectified Linear Unit (ReLU).** An activation function commonly used in DNNs. Defined as max(0, X). This function
is used as it is computationally cheap and avoids problems of vanishing gradients.
**Recurrent network.** A neural network which does pass the output from nodes of the network back into the input of
others. Infinite recurrence is avoided by setting a specific number of iterations for the recurrence. These are often
depicted in diagrams as separate layers but the implementation is through internal recurrent connections.
**Regression.** A model which attempts to diagnose or predict an exact value by statistically relating example input
values to desired values.
**Relevance vector machine.** A sparse Bayesian model utilising the kernel trick in similar fashion to a support vector
machine.
**Representation error.** Error which is introduced due to the inexactness of representing the real world in the model
belief state. Examples may include topography smoothing, point-to-grid translations, model grid distortions near the
poles, or the exclusion of physical characteristics which are not primary to the model.
**Residual neural network (ResNet).** A very influential and innovative convolutional NN architecture which uses a
similar concept to gradient boosting. Each layer of the deep network is taken to predict the residual error from the
previous layers, with skip-connections from earlier layers allowing the training to occur without the issue of vanishing
gradients.
**Sample.** A single training example (e.g. a row of data).
**Scale invariance.** A feature of a system, problem or model which means the results and behaviour are the same at any
scale (e.g., the behaviour does not change if the inputs are multiplied by a common factor).
**Scikit-learn.** A popular Python library for machine learning which extends the SciPy framework.
**Sharding.** Refers to dividing the training of a neural network across multiple GPUs or nodes. This can be done using
data sharding, whereby each GPU or node trains on a subset of the data to allow training parallelism, or model sharding
where a single model is partitioned across multiple GPUs to allow a larger neural network than could be allocated in
memory on a single GPU. One example could be assigning a small number neural network layers to each GPU which
could then work in sequence to operate on a very large network.
**(Stochastic) Gradient descent.** An algorithm by which a neural network is trained using increasingly fine-scale
adjustments to optimise the accuracy of network prediction. Utilised to find the local minimum of a differentiable
function.
**Supervised learning.** Machine learning is considered 'supervised' when the data is labelled according to a category
or target value. Classification data have an explicit labelled category. Regression data have an explicit value which is
being predicted for.
**Support vector machine.** A classification model based on finding a hyperplane to separate data utilising the kernel
trick.
**Tensor.** Can be considered as a dense multi-dimensional array or matrix.
**Tensorflow.** A widely adopted framework for neural networks in Python.
**Test/train/validate split.** Available data is split into three portions. The training data is evaluated and used to update
model weights. Validation data is evaluated during training and may be used for hyper-parameter search or to guide
the researcher. Test data is independent (typically well-curated) data used for gold standard evaluation. In reality,
validation data is sometimes used as test data, but this is not good practice. There are many considerations for
test/train/validate splitting, such as statistical independence, representation of all classes, and bias. It is important to
consider what the model is generalising "from" and "to", and ensuring appropriate examples are present in the training
data and appropriate examples are reserved for validation and test.
**Token.** Tokenisation the process of mapping a symbolic or categorical sequence to a numerical representation which
is suited to a sequence-based machine learning model. Commonly, a vector representation will be utilised for the token
form. In language processing, either characters or words may be represented as tokens depending on the approach.
**Top Hat function.** A filter or function which has a rectangular shape resembling the cross-section of a top hat. One
of the simplest functions used for convolutional operations, it can be defined as one constant value in a given bounded
range, and another smaller constant value outside that range.
**TPU.** Tensor Processing Unit. A hardware device specialised for artificial intelligence and machine learning
applications, in particular neural network operations.
**Training batch (or simply batch).** Multiple definitions apply and the use the term has evolved over time. Originally
used in the context of learning from offline or saved historical data as opposed to online or realtime novel data. In this
definition, the training batch is the saved data and refers to the whole training set. For example, a robot exploring a
new environment in real-time must use an online learning technique and could not utilise batch training to map the
unseen terrain. In more recent use, particularly in the areas of neural network learning, the offline saved data may be
split into one or more batches (subsets). If one batch (the batch is the entire training set) is used, the aggregate errors
for the entire training set are used to update the model weights and biases, and the learning algorithm is called batch
gradient descent. If each example is presented individually, this is called online training (even when historical saved
data is being used), the weights and biases are updated for from each individual example, and the algorithm used is
stochastic gradient descent. If the data is divided into multiple batches, this is often referred to equivalently as mini
batches. The weights and biases are aggregated over each mini batch. This is the most common contemporary
approach, as it reduces overfitting and is a good balance of training accuracy, avoiding local minima, and
computational efficiency.
**Transfer learning.** The process of training a model first on a related problem, and then conducting further training
on a more specific problem. Examples could be training a model first in one geographical region and then in another;
or training first at a low resolution then subsequently at a high resolution. This is frequently done to reduce training
computation cost for similar problems by re-using the trained weights from a well-performing source model, or to
overcome a problem of limited data availability by using multiple data sources.
**Transformer network.** A token-sequence architecture which is capable of handling long-range dependencies.
Initially applied to language processing, it has found effective application in image processing as an alternative to
convolutional architectures.
**Translation invariance.** A feature of a system, problem or model which means the results and behaviour are the same
after any spatial translation (i.e., the behaviour does not change if the inputs are shifted spatially to a new location).
**U-Net.** A type of convolutional neural network developed for biomedical image segmentation which has found broad
application. In the contracting part of the network spatial information is reduced while feature information is increased.
In the expanding part of the network, feature information is used to inform high-resolution segmentation. The name
derives from the diagrammatic shape of the network forming a "U".
**Unsupervised learning.** Machine learning is considered 'unsupervised' when data is unlabelled. Examples include
clustering, association and dimensionality reduction.
**Vanishing Gradient.** At the extremes, nonlinear functions used to calculate gradients can result in gradient values
which are effectively zero. These small or zero values, once present in the weights and biases of a neural network, can
entirely suppress information which would in fact be useful, and result in a local minima from which training cannot
recover. This is particularly relevant to long token-series when long-distance connections are relevant. A variety of
techniques including alternative activation functions, training weight decay, skip connections and attention
mechanisms may each or all be utilised to ameliorate this issue.
**Weights and biases.** The parameter values for each neuron which represent the weighting factors to apply to the input
values, plus an overall bias value for the node.
**XGBoost.** A popular Python library for gradient boosted decision trees.

**Appendix A: Table Summary of Model Architectures cited in this paper.**
This table includes all references from this review except for: seminal ML papers that are on new ML methods (e.g., foundational
ML papers), review papers, any paper cited that concerns a topic which is out of scope (e.g., nowcasting), and any other paper
which does not present a new method directly applicable to weather and climate modelling.

| Author(s) | Year | Category | Approach |
|---|---|---|---|
| Ackmann et al | 2020 | Fully connected NN | Preconditioner |
| Alemohammad et al | 2017 | Fully connected NN | Variable estimation |
| Andersson et al | 2021 | Convolutional NN | Prediction |
| Arcomano et al | 2022 | Reservoir computing | Alongside-model bias corrector |
| Atkinson | 2020 | Baysean type NN | PDE solver |
| Bar-Sinai | 2019 | Convolutional NN | PDE solver |
| Battaglia et al | 2018 | Graph NN | Method paper |
| Beucler et al | 2019 | Physics Informed NN | Convective paramterisation |
| Beucler et al | 2021 | Physics Informed NN | Convective paramterisation |
| Bhattacharya et al | 2021 | Fully connected NN | PDE solver |
| Bi et al | 2022 | Mixed/Custom NN | Pure ML atmospheric model |
| Bihlo & Popovych | 2022 | Physics Informed NN | PDE solver |
| Bolton and Zanna | 2019 | Convolutional NN | Parametrization |
| Brenowitz & Bretherton | 2018 | Fully connected NN | Parametrization |

| | | | |
|---|---|---|---|
| Brenowitz & Bretherton | 2019 | Fully connected NN | Parametrization |
| Brenowitz et al. | 2020 | Fully connected NN | Parametrization |
| Brenowitz et al | 2020 | Decision tree-based, Fully connected NN | ML model intercomaprison |
| Brenowitz et al | 2022 | Recurrent NN | Parametrization |
| Chaney et al | 2016 | Decision tree-based | Interpolation |
| Chantry et al | 2021 | Fully connected NN | Parametrization |
| Chattopadhyay et al | 2020 | Fully connected NN, Recurrent NN | Super parametrization |
| Chevallier et al | 1998 | Fully connected NN | Parametrization |
| Chi & Kim | 2017 | Fully connected NN, Recurrent NN | Prediction |
| Clare et al | 2021 | ResNet | Emulation (probabilistic) |
| Dagon et al | 2020 | Fully connected NN | Emulation |
| de Bézenac et al | 2017 | GAN | Prediction, model evaluation |
| Deuben and Bauer | 2018 | Fully connected NN | Replacement |
| Flora et al | 2022 | Decision tree-based, Logistic regression | Asessment of explainability techniques |
| Fuhg et al | 2022 | Physics Informed NN | PDE solver |
| Gagne et al | 2019 | Decision tree-based | Parametrization |
| Gagne et al | 2020 | GAN | Parametrization (probabilistic) |
| Gagne et al | 2020 | GAN, Fully connected NN | Parametrization |
| George et al | 2008 | Mixed/Custom non-NN | Preconditioner |
| Gettelman et al | 2021 | Fully connected NN | Emulation |
| Ham et al | 2019 | Convolutional NN | Prediction |
| Ham et al | 2021 | Convolutional NN | Prediction |
| Han et al | 2020 | ResNet | Parametrization |
| Harder et al | 2022 | Fully connected NN | Emulation |
| He et al | 2022 | Decision tree-based | Parametrization |
| Holloway & Chen | 2007 | Fully connected NN | Preconditioner and PDE solver selection |
| Horvat & Roach | 2022 | Fully connected NN | Parametrization |
| Hu et al | 2023 | Mixed/Custom NN | Pure ML atmospheric model |
| Huang et al | 2016 | SVM | Preconditioner |
| Kapp-Schwoerer et al | 2020 | Convolutional NN | Semantic segmentation |
| Karunasinghe & Liong | 2006 | Fully connected NN | Chaotic timeseries prediction |
| Keisler | 2022 | Graph NN | Replacement |
| Kim et al | 2022 | Mixed/Custom NN | Prediction |
| Kochkov et al | 2021 | Convolutional NN | PDE solver |
| Krasnopolsky et al | 2002 | Fully connected NN | Emulation |
| Krasnopolsky et al | 2005 | Fully connected NN | Emulation |
| Krasnopolsky | 2013 | Fully connected NN | Parametrization (probabilistic) |

| | | | |
|---|---|---|---|
| Kuefler & Chen | 2008 | Mixed/Custom non-NN | Linear system solver |
| Ladický et al | 2015 | Decision tree-based | PDE solver |
| Lam et al | 2022 | Mixed/Custom NN | Pure ML atmospheric model |
| Lanthaler et al | 2022 | Neural Operator | PDE solver |
| Leufen & Schadler | 2019 | Fully connected NN | Paramterization |
| Li et al | 2020 | Graph NN | PDE solver |
| Li et al | 2020 | Neural Operator | PDE solver |
| Li et al | 2020 | Neural Operator | PDE solver |
| Lopez-Gomez et al | 2023 | Convolutional NN | Prediction |
| Lu et al | 2020 | Neural Operator | PDE solver |
| Meyer et al | 2022 | Fully connected NN | Emulation |
| Moishin et al | 2021 | Convolutional Recurrent NN | Prediction |
| Mooers et al | 2021 | Fully connected NN | Emulation |
| Mudigonda et al | 2017 | Mixed/Custom NN | Object detection |
| Nelsen & Stuart | 2021 | Random Feature Model | PDE solver |
| Nguyen et al | 2023 | Mixed/Custom NN | Pure ML atmospheric model |
| O'Brien et al | 2020 | Baysean model | Object detection |
| O'Gorman & Dwyer | 2018 | Decision tree-based | Emulation |
| O'Leary et al | 2022 | Fully connected NN | PDE solver |
| Ott et al | 2020 | Fully connected NN | Emulation |
| Pan et al | 2020 | Decision tree-based | Paramterisation |
| Patel et al | 2021 | Neural Operator | PDE solver |
| Pathak et al | 2022 | Mixed/Custom NN | Pure ML atmospheric model |
| Peairs & Chen | 2011 | Mixed/Custom non-NN | PDE solver |
| Pelissier et al | 2020 | Mixed/Custom non-NN | Hybrid model corrector |
| Prabhat et al | 2021 | Convolutional NN | Object detection |
| Psaros et al | 2023 | Neural Operator, Physics Informed NN | PDE solver |
| Rasp | 2020 | Fully connected NN | Emulation |
| Rasp et al | 2018 | Fully connected NN | Emulation |
| Rasp et al | 2020 | Fully connected NN, Linear regression | Pure ML atmospheric model |
| Rasp & Thuerey | 2021 | ResNet | Pure ML atmospheric model |
| Rizzuti et al | 2019 | Convolutional NN | NN based corrector step in PDE solver |
| Rosier et al | 2023 | Mixed/Custom NN | Prediction |
| Ross et al. | 2023 | Genetic programming, Linear regression, Convolutional NN | Intercomparison of methods to learn paramterisations from data |
| Rupe et al | 2023 | Mixed/Custom non-NN | Object detection |
| Sawada | 2020 | Regression | Emulation |
| Scher | 2018 | Convolutional NN | Emulation |

| | | | |
|---|---|---|---|
| Scher and Messori | 2019 | Convolutional NN | Emulation |
| Taylor & Feng | 2022 | Convolutional NN | Prediction |
| Tompson et al | 2017 | Convolutional NN | PDE solver |
| Toms et al | 2020 | Fully connected NN | NN interpretability |
| Ukkonen & Mäkelä | 2019 | Decision tree-based, Logistic Regression, Fully connected NN | Paramterisation |
| Ukkonen et al | 2020 | Fully connected NN | Emulation |
| Vlachas et al | 2018 | Recurrent NN | Pure ML baseline model |
| Wang et al | 2021 | Neural Operator | PDE solver |
| Wang et al | 2022 | ResNet | Parametrization |
| Wang et al | 2022 | Physics Informed NN | PDE solver |
| Watt-Meyer et al | 2021 | Decision tree-based | Nudging |
| Watson-Parris et al | 2022 | Gaussian Process, Decision tree-based, Mixed/Custom NN | Pure ML baseline model |
| Weyn et al | 2019 | Convolutional NN | Pure ML atmospheric model |
| Weyn et al | 2020 | Convolutional NN | Pure ML atmospheric model |
| Weyn et al | 2021 | Convolutional NN | Pure ML atmospheric model |
| Wikner et al | 2020 | Reservoir computing | Alongside-model bias corrector |
| Wu & Xiu | 2020 | ResNet | Learning PDE operators |
| Yamada et al | 2018 | Convolutional NN | Preconditioner |
| Yang et al | 2016 | Fully connected NN | PDE solver |
| Yeo et al | 2021 | Recurrent NN | Dynamical system simulation |
| Yuval & O'Gorman | 2020 | Decision tree-based | Emulation |
| Yuval et al | 2021 | Fully connected NN | Emulation |
| Zanna and Bolton | 2020 | Convolutional NN, Relevance vector machine | Parametrization and equation discovery |
| Zhao et al | 2019 | Fully connected NN | Paramterisation |
| Zhao et al | 2019 | Physics Informed NN | Paramterisation |
| Zhong et al | 2023 | Fully connected NN, Recurrent NN | Emulation |



**Code Availability**


No code was used in the preparation of this review.

**Data Availability**


No data was processed in the preparation of this review except for the list of ML model types by cited paper, which
is provided in the appendix.
**Author Contribution**
COdBD researched and wrote Sections 3, 4, 5, 6 and 7, and provided review of sections 8, 10, and the glossary. TL
researched and wrote sections 8, 10, and the glossary, and provided review of sections 3, 4, 5, 6, and 7. COdBD and
TL researched and co-wrote sections 1, 2, 9, 11, 12, and the Appendix.
**Competing Interests**
The authors declare that they have no conflict of interest.
**Acknowledgements**
The authors would like to thank Bethan White, Harrison Cook, Tom Dunstan and Karina Williams for their very
helpful reviews of early versions of this manuscript. We also would like to wholeheartedly thank the referees for their
extremely helpful, positive and well considered feedback and suggestions. Their input has greatly improved this
review. Finally, we would like to acknowledge and thank the people who contacted us with comments, suggestions,
and advice on the preprint versions of this review. All of the input was valuable, and greatly appreciated.

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
