# Peer review of "Machine Learning for numerical weather and climate modelling: 1 a review 2"

_EGUsphere, 2023_

## Author Comment (AC1)

We would like to thank the referee for their very helpful and constructive feedback. We feel that their advice has very much improved the quality of the review, especially relating to the Introduction and Subsection 5.3.

We have responded inline to the referee's comments in blue font below.

**Anonymous Referee #1**

**General comments:**

The authors review the application of Machine Learning (ML) techniques to weather and climate modelling with an emphasis on historical and current developments. A glossary of commonly used terms and basic introductions to some concepts are provided. An in-depth exchange of knowledge between the ML and geoscientific modelling communities could be immensely beneficial and reviews like this can be an important step to facilitate this exchange.

As far as I am able to judge, the authors do a great job at covering a wide range of relevant publications and explaining the questions tackled in many of these applications. In terms of presentation, the language is concise and the paper is enjoyable to read. Including tabular or schematic representations of ML concepts and/or the discussed applications could further improve the visual appeal of the paper. A stronger narrative thread linking the different subsections and applications would preempt the impression of reading through a long list of papers - although this may be unavoidable given the scope of the reviewed works.

We thank the referee for their kind words. We did indeed endeavor to have a narrative thread through the sections, however as the referee notes, this was somewhat challenging to do given the wide scope of the review. We have reworded the final paragraph of the introduction (summarizing the remaining sections) to try to clarify the logic of the order of the sections somewhat.

We also agree with the referee that some figures or tables would add to the visual appeal of the review. We will explore some relevant visualizations and add them in if we feel they increase the visual appeal and information content of the review.

My primary concern about the paper in its current form is its utility to aid researchers in the development of better geoscientific models. Due to the wide range of works that are being discussed, many concepts and models are only touched upon in brief, without further elaboration of the underlying principles and connections between different applications. References to methodological works that could support future model development are only sparsely included in the main text or the glossary. In my opinion, incorporating suitable methodological references into the glossary and introductory sections could greatly strengthen the paper!

This is a very good suggestion. We have added more explanation of a selection of good foundational papers and books in Section 2:

"Suggested starting points for interested readers, including guidance on the utility of different model architectures and algorithms, and the connections between different applications and approaches, are as follows:

- Hsieh (2023) provides a thorough textbook on environmental data science including statistics and machine learning
- Chase et al (2022a, 2022b) provide an introduction to various machine learning algorithms with worked examples in a tutorial format and an excellent on-ramp to ML for weather and climate modelling
- Russell & Norvig (2021) provide a comprehensive book regarding artificial intelligence in general
- Goodfellow et al. (2016) provide a well-regarded book on deep learning theory and modern practise
- Hastie et al. (2009) provide a book on statistics and machine learning theory"

While we think that a review which focused on the relative strengths and weaknesses of each ML algorithm and architecture would be of great value, it is not the primary focus of this review. This review seeks to provide an overview of the major developments in the research around ML for weather and climate modeling. To also include an exploration of the methodological strengths and weaknesses of different architectures and algorithms would, we believe, significantly increase the size of an already very long manuscript. We have thus left this to future work, and would view this review as being complimentary to one exploring the methodological aspects of ML for weather and climate modelling in more detail.

**Specific comments:**

L20 - Isn't there an ongoing research effort to extend numerical models to utilise GPU hardware?

This is true, however it is still not an easy task. We have amended this sentence to clarify that it is doable but difficult:

"These numerical weather and climate forecasts are computationally costly and are not easy to implement on specialized compute resources such as GPUs (although there are efforts underway to do so, for example in LFRic (Adams et al. 2019))."

The fact work is underway to make numerical models able to run on GPUs is also acknowledged in Section 7.4

L24 - What about improvements in subgrid parameterizations due to better process understanding?

We have amended this sentence to include this: "An additional pathway to improve skill is to improve the understanding and representation of sub grid-scale processes, however this is again a potentially computationally costly exercise."

L65/66 - Maybe include a reference? (e.g. McGovern et al 2019 [1])

Good suggestion – some references added: "(e.g., McGovern et al., 2019; Toms et al., 2020; Samek et al., 2021)"

McGovern, A., Lagerquist, R., Gagne, D. J., Jergensen, G. E., Elmore, K. L., Homeyer, C. R., & Smith, T. (2019). Making the black box more transparent: Understanding the physical implications of machine learning. Bulletin of the American Meteorological Society, 100(11), 2175-2199.

Samek, W., Montavon, G., Lapuschkin, S., Anders, C. J., & Müller, K. R. (2021). Explaining deep neural networks and beyond: A review of methods and applications. Proceedings of the IEEE, 109(3), 247-278.

Toms, B. A., Barnes, E. A., & Ebert-Uphoff, I. (2020). Physically interpretable neural networks for the geosciences: Applications to earth system variability. Journal of Advances in Modeling Earth Systems, 12(9), e2019MS002002.

L104 - Very debatable if this is a necessary requirement for e.g. a weather prediction model?

It's not necessary if other items in the list are satisfied, but the list specifies "one or more of". We would suggest that if the model didn't provide any of the other benefits in the list, it would have to at least provide the last item in the list; insight into physical processes not provided by current numerical models or theory

L116ff - A narrative thread linking these subsections would be much appreciated!

We have reworded this paragraph to try to illustrate more of a narrative thread through the sections:

"The remainder of this review is structured as follows: In Section 2 a quick introduction to ML is provided, before the application of ML in weather and climate modelling is explored in the following five sections. Firstly, ML use in sub-grid parametrization and emulation, along with tools and challenges specific to this domain, are covered in Section 3. Zooming out from sub-grid scale to processes resolved on the model grid, in Section 4 the application of ML for the partial differential equations governing fluid flow is reviewed. Expanding scope yet again to consider the entire system, the use of ML for full model replacement or emulation is reviewed in Section 5. In Section 6 the growing field of physics constrained ML models is introduced, and in Section 7 a number of topics tangential to the main focus of this review are briefly mentioned. Setting the work covered in the previous sections in a broader context, a review of the history of, and progress in, ML outside of the fields of weather and climate science is presented in Section 8. In Section 9 some practical considerations for the integration of ML innovations into operational and climate models are discussed, and finally a summary is presented in Section 10. A Glossary of Terms is provided after the final Section to aid the reader in their understanding of key concepts and words."

L128 - Could it be more useful to briefly discuss the utility of the individual references rather than providing a large list?

As was already mentioned above, we have expanded briefly on the topics covered by each of the references to help guide the reader to the references most useful to them:

"Suggested starting points for interested readers, including guidance on the utility of different model architectures and algorithms, and the connections between different applications and approaches, are as follows:

- Hsieh (2023) provides a thorough textbook on environmental data science including statistics and machine learning
- Chase et al (2022a, 2022b) provide an introduction to various machine learning algorithms with worked examples in a tutorial format and an excellent on-ramp to ML for weather and climate modelling
- Russell & Norvig (2021) provide a comprehensive book regarding artificial intelligence in general
- Goodfellow et al. (2016) provide a well-regarded book on deep learning theory and modern practise
- Hastie et al. (2009) provide a book on statistics and machine learning theory"

L136 - Debatable, as recent trends in ML point strongly in the opposite direction (i.e. larger homogenised models).

We don't necessarily agree with the assertion of the referee, however we do acknowledge that it is a sufficiently nuanced and debatable premise (both in terms of our claim and the counterclaim made by the referee) that it is not suited to a single sentence at the end of the paragraph. It is not an essential point to resolve for this review, so in the interests of brevity we have simply removed the sentence.

L145 - Debatable as emphasis shifts towards self-supervised learning and better training regimes rather than architectural developments!

We do not entirely agree with the referee on this point, however we do acknowledge that alongside architectural/algorithmic improvements (e.g., Earthformer), gains have been made through improved training regimes (e.g. FengWu and assorted parametrization scheme examples). We also realize that this sentence was somewhat ambiguous in that it wasn't clear as to whether it was referring to ML in the context of weather and climate modelling, or ML in a more general context. We do acknowledge that in a boarder context there is a relatively greater focus on unsupervised learning currently. We have amended this sentence to acknowledge the other areas where research is currently focused in the weather and climate modelling context:

"A major current focus of ML research in the context of weather and climate modelling is new NN-based architectures and algorithms, and improved training regimes."

L156 - It could be important to emphasise that NN are known to interpolate within the training envelope and may not generalise well outside it (in contrast to physical laws).

We agree that this is important to point out. We have mentioned it later on in discussing applications of ML (especially for parametrization schemes) but agree that it is worth mentioning here too. We have amended this sentence to state: "NNs can therefore theoretically be candidates for accurate modelling of physical processes, although in practise they cannot always reliably predict beyond their training envelope and as such may not generalize to new regimes."

L163 - Sigmoid is a highly uncommon and suboptimal choice of activation function compared to ReLU! (ReLU is also missing from the glossary despite its ubiquity in modern models).

We have amended the text to clarify the cases in which sigmoid functions are used and have added a comment on cases where other activation functions are used:

"A commonly used activation function for a single neuron is the sigmoid function, which helpfully compresses the range between 0 and 1 while allowing a nonlinear response."

"Larger networks make more use of linear activations and may utilize heterogenous activation function choices at different layers."

ReLU has also been added to the glossary: "Rectified Linear Unit (ReLU). An activation function commonly used in DNNs. Defined as max(0, X). This function is used as it is computationally cheap and avoids problems of vanishing gradients."

L179 - Why are Token-sequence and Transformer models listed separately? I don't see the justification for this classification introduced as is.

There are token-sequence architectures that are not transformers. Many of the token-sequence architectures don't do dimensionality reduction, distinguishing them from transformers in that way. We have amended the list to clarify this and add some extra categories:

- "Small, fully-connected networks, which are less commonly featured in recent publications but are still effective for many tasks and are still being applied and may well be encountered in practice
- Convolutional† architectures, first applied to image content recognition, which match the connectome of the network to the fine structure of images in hierarchical fashion to learn to recognize high-level objects in images
- Recurrent token-sequence architectures, first applied to natural language processing, generation and translation; applicable to any time-series problem. Now also applied to image and video applications, and mixed-mode applications such as text-to-image or text-to-video
- Transformer architectures†, based on the attention mechanism† to provide a non-recurrent architecture which can be trained using parallelized training strategies. This allows larger models to be trained. Originally developed for sequence prediction and extended to image processed through vision transformer architectures."

L521 - ConvLSTM were introduced in 2015 also in the context of nowcasting, including a reference would be appropriate [2].

We were actually not aware that this was the origin of ConvLSTMs – a significant oversight on our part. Thank you for drawing this to our attention! We have added a sentence explaining the origins of ConvLSTMs and the reference you recommended:

"Convolutional LSTMs (ConvLSTMs), which combine convolutional layers with an LSTM mechanism, were introduced in the meteorological domain by Shi et al. (2015) for precipitation nowcasting. They have since seen wide adoption in other areas (e.g., Yuan et al., 2018; Moishin et al., 2021; Kelotra & Pandey, 2020). Their success in other domains suggests that revisiting their utility for weather and climate modelling could be worthwhile."

Shi, X., Chen, Z., Wang, H., Yeung, D. Y., Wong, W. K., & Woo, W. C. (2015). Convolutional LSTM network: A machine learning approach for precipitation nowcasting. Advances in neural information processing systems, 28.

Moishin, M., Deo, R. C., Prasad, R., Raj, N., & Abdulla, S. (2021). Designing deep-based learning flood forecast model with ConvLSTM hybrid algorithm. IEEE Access, 9, 50982-50993.

Yuan, Z., Zhou, X., & Yang, T. (2018, July). Hetero-convlstm: A deep learning approach to traffic accident prediction on heterogeneous spatio-temporal data. In Proceedings of the 24th ACM SIGKDD International Conference on Knowledge Discovery & Data Mining (pp. 984-992).

Kelotra, A., & Pandey, P. (2020). Stock market prediction using optimized deep-convlstm model. Big Data, 8(1), 5-24.

L537 - Why is Sonderby et al discussed if nowcasting is supposedly omitted (L515)? Why not Espeholt et al 2021? Why is this not discussed in the context of probabilistic models?

This is a reasonable point, and we agree that Sonderby is out of scope. We have removed the sentences describing this paper.

L560 - A background reference to GNN either here or in the glossary could be beneficial! (e.g. Battaglia et al 2018 [3])

This is a good suggestion – thank you. We have added your suggested reference to the Glossary, along with Scarselli et al. (2008) and Kipf & Welling (2016).

Kipf, T. N., & Welling, M. (2016). Semi-supervised classification with graph convolutional networks. arXiv preprint arXiv:1609.02907.

Scarselli, F., Gori, M., Tsoi, A. C., Hagenbuchner, M., & Monfardini, G. (2008). The graph neural network model. IEEE transactions on neural networks, 20(1), 61-80.

L581 - I would strongly object to treating the models in this section (excluding Clare et al) as probabilistic in contrast to the ones in the previous section! These models are fundamentally deterministic, in contrast to e.g. generative models such as Ravuri et al or true probabilistic models like Sonderby et al. Discussing the different types of ensembling used in these models could be valuable on its own (also referring to Scher et al [4]).

We understand and acknowledge the basis for your objection – we were using the generation of an ensemble as a proxy for probabilistic modelling because that is the most common method for producing probabilistic outputs from physics based models in a forecasting setting, but by doing so we are definitely showing our biases, and we can see how that is an unreasonable division to make when there are many examples of intrinsically probabilistic ML models, some of which were included in the previous sections. We also agree that there is value in examining the question of ensemble ML models in more detail in any case.

We have shifted our description of Clare et al into the previous sections where it fits better in the narrative, and have significantly re-written section 5.3 as an examination of ML models used to generate ensemble predictions.

L661 - A large part of the affordable training is the use of much lower resolution and not due to the architecture! (1.4deg vs 0.25)

We acknowledge that this was overstating the fact, and wasn't making the main point well. We have amended the sentence to be clearer:

"Furthermore, ClimaX used novel encoding and aggregation blocks in its architecture to enable greater flexibility in the types of variables used for training, and to reduce training costs when a large number of different input variables were used."

L663 - Missing several extensions of WeatherBench (e.g. WeatherBenchProbability, RainBench) and ClimateBench.

Thank you for pointing this out – we became aware of these omissions soon after submission and have now added them in:

"Since the publication of WeatherBench, more benchmark datasets tailored to other domains have been created, including RainBench (de Witt et al., 2020), WeatherBench Probability (Garg et al., 2022), and ClimateBench (Watson-Parris et al., 2022)."

Garg, S., Rasp, S., & Thuerey, N. (2022). WeatherBench Probability: A benchmark dataset for probabilistic medium-range weather forecasting along with deep learning baseline models. arXiv preprint arXiv:2205.00865.

de Witt, C. S., Tong, C., Zantedeschi, V., De Martini, D., Kalaitzis, F., Chantry, M., ... & Bilinski, P. (2020). RainBench: towards global precipitation forecasting from satellite imagery. arXiv preprint arXiv:2012.09670.

Watson-Parris, D., Rao, Y., Olivié, D., Seland, Ø., Nowack, P., Camps-Valls, G., ... & Roesch, C. (2022). ClimateBench v1. 0: A Benchmark for Data-Driven Climate Projections. Journal of Advances in Modeling Earth Systems, 14(10), e2021MS002954.

L981 - The activation function is applied elementwise to the result of a matrix multiplication and does not incorporate multiplication or bias addition by definition.

We have amended our definition to be more clear:

"Activation Function. The function which produces a neuron's outputs given its inputs. Commonly, this includes a learned bias term which is added to the data inputs before evaluation with a single function to produce the output value. Examples of the functions used include linear, sigmoid and tanh."

L995 - Calling it a "complex" mechanism is not necessary. Typical Attention simply computes a dot product between vectors.

This is fair enough – we have removed the word complex.

L1007 - Normalisation plays an essential role in modern NN and probably deserves its own glossary term. It does not need to be performed over the batch (i.e. LayerNorm)

This was an erroneous omission. We have added a definition of normalisation:

"Normalisation. A technique applied in many areas of mathematics, science and statistics which is also very important to machine learning and neural networks. In a general sense, this refers to expressing values within a standard range. Very often, the range of expected values is mapped onto the range 0 to 1, to allow physical variables with different measurement units to be compared on equal scale. Such normalisation may be linear or nonlinear, according to a simple or more complex function, and either drawn from known physical limits or from the variation observed in the data itself."

L1028 - "Convolutional … sliding window" seems redundant.

Given the broad scope of possible readers of this review, we think there may be readers who would be familiar with the concept of a sliding window programmatically, but would not be familiar with the term convolution. Thus we think it is good to keep both, even though they are somewhat redundant. We have amended this sentence slightly to: " Convolutional neural network. A neural network architecture commonly applied to images which utilises a convolutional (spatially connected) kernel applied in a sliding window fashion with a narrow receptive field to encourage the network to generalise from fine scale structure to higher levels of abstraction."

L1037 - Not true in general! If the network is too thin it becomes highly unstable.

We have amended this definition: "Deep NN. A neural network with many layers. Deeper, thinner networks have generally been more popular in recent times than wider, shallower ones but this is not always the case (see e.g. https://arxiv.org/abs/1605.07146)."

**Technical corrections:**

DNN and NN are introduced as separate abbreviations, but the distinction is not kept consistent nor does it appear beneficial.

This is a reasonable observation – we have adopted a more consistent convention of just using NN.

**References:**

[1] McGovern, A., R. Lagerquist, D. John Gagne, G. E. Jergensen, K. L. Elmore, C. R. Homeyer, and T. Smith, 2019: Making the Black Box More Transparent: Understanding the Physical Implications of Machine Learning. Bull. Amer. Meteor. Soc., 100, 2175–2199, https://doi.org/10.1175/BAMS-D-18-0195.1.

[2] Shi, Xingjian, et al. "Convolutional LSTM network: A machine learning approach for precipitation nowcasting." Advances in neural information processing systems 28 (2015).

[3] Battaglia, Peter W., et al. "Relational inductive biases, deep learning, and graph networks." arXiv preprint arXiv:1806.01261 (2018).

[4] Scher, Sebastian, and Gabriele Messori. "Ensemble methods for neural network-based weather forecasts." Journal of Advances in Modeling Earth Systems 13.2 (2021).

---

## Author Comment (AC2)

We would like to thank the referee for their very helpful and constructive feedback. They have identified some areas where we are able to greatly improve the review. Their suggestions for figures and tables, and for future directions were particularly useful.

We have responded inline to the referee's comments in blue font below.

**Anonymous Referee #2**

In "Machine Learning for numerical weather and climate modelling: a review," de Burgh-Day and Leeuwenburg provide a review–aimed at weather/climate model developers–of machine learning itself, the history of its application in weather/climate modeling, and contemporary uses (and challenges) including: parameterization replacement, coarse-graining, superparameterization, fluid dynamics solvers, and others.

Overall the review is written clearly and written well for the intended audience, it is comprehensive with respect to the ML literature associated with weather & climate modeling, and it correctly and adequately summarizes the relevant literature. Overall I think this will be an invaluable addition to the literature, complementing some other recent reviews in the ML+weather/climate literature.

We thank the referee for their kind words. We are heartened that they think this review will be of value and have found their feedback and suggestions to be immensely useful. They identified some gaps in the review which we will work to address before we upload our revised version.

The three main places where, in my opinion, the review could use some improvement are: (1) addition of graphics/diagrams/tables/pseudocode/anything-to-please-break-up-the-text, (2) filling some gaps in the review, and (3) more synthesis. I also have some other minor improvements to suggest. Detailed comments follow.

*Note that two students also read this paper and provided input on the review.

Thank you for these suggestions. We have responded to each of them inline below.

**Major feedback**

**Adding figures, diagrams, tables, pseudocode, etc.**

The stated goal of the manuscript, "to provide a primer for researchers and model developers to rapidly familiarize and update themselves with the world of ML in the context of weather and climate models," would be better-served if visual aids of some form were added to the manuscript, particularly in Sections 2 and 8–10, and possibly also elsewhere.

I'd also add that the current version of the paper is a lot of text without any interruption; I found that this made it difficult for me to hold my attention on the paper.

We agree that this would improve the visual appeal of the review, and note that the other referee made the same suggestion. We have identified some areas where useful visualizations can be added, although we feel that there may be a limit to the number of visualizations we can add that would be worthwhile and wouldn't just lead to the review becoming even longer. Thank you for the suggestions of visualizations for different sections – this is a very helpful guide for us to use in what we could add that would be of value.

Some specific suggestions for figures/diagrams/etc. follow.

**Section 2** Readers who are totally unfamiliar with machine learning may find it challenging to derive meaning from the text-only descriptions given in Section 2. I'm not necessarily advocating for yet another elementary neural network diagram, like one could find on wikipedia, but rather something that will help this specific audience–model developers–form a reasonably good mental model of neural networks, decision trees, and the various architectures of them. This could be in the form of a diagram, but this audience also might find it simpler to digest some pseudocode: e.g., pseudocode describing a neuron as a function, or pseudocode describing how a convolutional layer works. But then again, maybe a diagram would be better.

A visual aid along the lines of what you describe is a good idea – thank you. We are developing a flowchart-style diagram which will provide readers with a framework for deciding which ML architectures and algorithms would be good candidates to try for a given problem. We are hoping to include in it considerations such as which of the simpler but often quite effective algorithms they should try initially, and then which of the more complex ML algorithms and architectures they may want to move on to, based on the nature of their problem (e.g. is it temporal, spatial, or spatiotemporal in nature?).

We have also included a brief table showing the strengths and weaknesses of the ML algorithms and architectures mentioned in this review, with a reference to a paper with another similar table in it as well.

**Sections 3–7** It could be useful to have a figure somewhere that gives a pie chart of the various ML/weather+climate modeling topics (e.g., how many papers, relatively, are in each of the categories outlined in the various sections.) Also, if any of the papers here are the authors's own work, then perhaps it wouldn't be to difficult to add a variant of a figure that already exists in the literature.

Also if any of the sources are published with a Creative Commons license, like many in GMD, then it should be acceptable to actually take figures from those papers as long as they are properly attributed following the guidelines. This could be a really simple way to break up the text and help the readers get a deeper glimpse into the work that has been done.

We have added a figure showing the breakdown of ML architectures in the papers referenced in this review (we had actually already started working on a figure like this before this feedback was posted, so it was pleasing to see this as a suggested visualization!)

**Section 8** It would be great to add a graphically-rich timeline that complements the list in section 8.1. I could imagine future authors using that timeline in ML presentations, which would be a great way to get free advertising for this paper.

We think this is a really good idea, but are unsure whether we will be able to make such a timeline that will be sufficiently compact. We'll give it a go!

**Section 9** Maybe I don't have an idea for a figure in this section after all.

**Section 10** Consider adding a table of new advances in ML that haven't yet been employed in weather/climate modeling, but that may be useful. Also/alternatively, consider adding a figure that

somehow communicates the promising new directions. It could be something as simple as a PowerPoint SmartArt that simply adds some graphical elements to highlight the text.

We have added a new section (section 11) titled "Future research directions" which summarises some potential focuses for future development and research, including identifying ML advances which haven't yet been tested for weather and climate modelling applications. We are leaning away from including a figure, because there is too much detail to fit in a figure, but instead we plan to include a table summarizing possible future research directions.

**Some gaps**

We have deliberately kept the scope of the review very narrow, since without doing so we would have risked this manuscript turning into a book. We have addressed each of the suggested additions inline below.

**Data Assimilation** I've come across some literature on ML and data assimilation (some even in GMD), but the authors don't discuss this at all here. Given that data assimilation is a critical component of weather modeling efforts, it would be a shame to overlook this. I suggest that the authors survey the literature on this.

We decided that use of ML in data assimilation is too large a topic to cover in this review – it deserves a review in its own right. We have instead focused on the step that comes afterwards – the prediction of subsequent states after the DA is complete, specifically on weather, subseasonal and seasonal, to climate prediction timescales. This is the same reason as the one that led us to not include nowcasting or to delve deeply into the use of machine learning for climate projections. Essentially, including these topics would expand the scope and length of this review beyond what is reasonable. We do, however, strongly agree with the referee that there is value in a review covering these topics and hope to see this emerge from the community in the future.

**Ice sheet modeling** Unless I overlooked it, this review doesn't discuss ice sheet modeling at all, which is an increasingly important component of CMIP-class models. A quick google scholar search on 'ice sheet model machine learning' turns up some apparently relevant results.

This is certainly an omission on our part – we will explore the literature in this space and add it into the review.

**Integrated assessment modeling / multi-sector dynamics** In the lifecycle of CMIP model efforts, generation of climate scenarios (like the SSPs) is a key step. This isn't discussed at all. There are at some efforts in this area that are worth mentioning, and I'd guess there are others, e.g:
https://www.osti.gov/biblio/1769796

For similar reasons to those stated above, we have made an explicit decision to not explore too deeply the topic of climate change projections and climate scenarios, and have instead stopped at multiyear and free-running simulations. We are aware that there is a very large body of literature on the application of ML to aspects of climate scenario modelling and as stated for previous topics, we feel that this topic is better suited to its own review. We do note however that the distinction between multiyear and free-running simulations, and climate projections/scenarios, was not made explicit, and the fact that one is kept in scope while the other isn't is not made clear.

We have added some text to the introduction to make this more clear:

"Additionally, here we consider climate modelling in the context of multiyear and free-running multidecadal simulations, but exclude the topic of ML for climate change projections, climate scenarios, and multi-sector dynamics. This is again in the interests of ensuring the scope of the review is manageable, rather than because these topics are not worthy of review. On the contrary, a review dedicated to the utility of machine learning in this area would be of enormous value to the community, but could not be done proper justice to here."

**AI Ethics** AI advances have ethical implications, and I think there might be some here too. It might be worth surveying some of the recent literature on ethics in AI, with a goal of summarizing the main ethical issues that come up with AI in general and what the implications of these ethical issues might be for AI in weather and climate model development.

This is a good point – the ethics of AI is indeed an important consideration. We don't feel that a full review of the ethical consideration of AI more generally is appropriate for this review, however, we do feel that it is very important for readers of this review to be made aware that there are ethical considerations associated with the use of AI. We have added a new section (section 10) discussing the ethical considerations of AI in its application to weather and climate modelling to the review, including references for readers interested in exploring the topic further.

**Synthesis**

In my perspective, the most impactful review articles are ones that (a) provide a comprehensive overview of the state of the literature (which this paper does quite well), and (b) synthesize what the authors have learned: even suggesting new directions that might not be immediately evident. The current version of the manuscript does great on (a) but does not do too much with respect to (b). The Conclusions section does this to some extent (e.g., "Nonetheless, there are still many challenges to overcome…This list provides a set of focus areas for future research efforts."), but the list focuses on challenges rather than promising new directions. The last paragraph starts to get at this with the sentence "Advances in the sophistication, complexity and efficiency of ML architectures are being heavily invested in…," but the manuscript then stops short of discussing these new advances or how they might point to new directions.

I recommend revamping the last section to focus on this synthesis aspect.

I'm not in the best position to give good suggestions here since I don't have as comprehensive of a knowledge of this literature as the reviewers, but after reading the paper, some untouched directions do come to mind:

- More exploration of foundation models. The authors note one recent example of a foundation model. The proliferation of foundation models in the last year (ChatGPT, for example) has this at the forefront of a lot of people's minds: what new research could contribute to the application/analysis/use of foundation models in weather and climate?
- Relatedly, it could be impactful to somehow fuse weather / climate code and data with GPT-like models. What sort of impact would it have if a model developer user could get insight from an AI model that's able to ingest and interpret high-dimensional data as well as code: e.g., "WxGPT,

why does the new change in commit 3efde6 result in a systematic cold bias in daytime maximum temperature forecasts?"

- Model emulation / tuning. There's some literature on groups using Gaussian process models to emulate climate models, where they use the emulators for quantifying uncertainty in tuning parameters and for finding optimal tunings; other ML methods could be useful here
- Model spinup: a major barrier to use of ultra-high-resolution coupled climate models is the time required to spin-up the slow components of the system like ocean and land ice; ML methods could potentially be useful here (e.g., for learning how to translate equilibrated states from a low-res model to a high-res model)
- 3D radiative transfer for high-res models: possibly replacing a full 3D radiative transfer code with an ML approximation, or perhaps using the climate model for 1D radiative transfer and using ML to model the expected differences in fluxes due to 3D effects
- Modeling full PDFs: cutting-edge models like FourCastNET essentially emulate what dynamical models do in that they provide deterministic (albeit presumably chaotic) states. What if instead they could be trained to output a PDF of states (e.g., emulating Fokker-Planck equations) rather than deterministic states? That would be something fundamentally new relative to existing model capabilities.

My main point here is that your review already has a lot of value in establishing what has already been done, and I think this paper will be more impactful if you increase the emphasis on what could plausibly be done that has not yet been touched. I recommend thinking 5-10 years into the future rather than just incremental advances based on what's been done. You have the unique opportunity to inspire others to try some radical new ideas.

Also, consider that this review will very likely be cited in workshop reports that inform funding agency priorities. Your last sentence states 'academic and operational agencies will need to continue to support research in this space;' giving specific ideas here could really have an impact.

This is a very good point and a good suggestion – thank you. We have modified the final section to provide more synthesis of the outcomes of the review. We are also adding a new section (section 11) to discuss future research directions.

Finally, I recommend also mentioning AI ethics in this last section. If you're going to inspire researchers to think radically, it would be responsible to also admonish people to always consider the ethical issues with as much mental effort as they do the technical issues. I can't help but quote from Jurassic Park here: "Your scientists were so preoccupied with whether or not they could, they didn't stop to think if they should."

Please see above for our thoughts on discussing AI ethics in this review. We agree that there is a strong ethical risk in the use of AI if it is not done with care, and we have added a new section (section 10) to emphasize this (as detailed above).

**Minor feedback**

The NVIDIA group has just put up a preprint of the newest version of FourCastNet, which allows them to perform year-length simulations: https://arxiv.org/abs/2306.03838

Thank you for drawing our attention to this. We are aware of a number of relevant publications which have come out since submitting this version of this review, for example the preprint for FengWu (https://arxiv.org/abs/2304.02948), for SwinRDM (https://arxiv.org/pdf/2306.03110.pdf), and several interesting papers making progress in the use of ML for atmospheric parameterization schemes (e.g. https://gmd.copernicus.org/articles/16/2355/2023/).

We will need to assess the latest additions to the literature since the initial submission of this review and decide whether it is worthwhile adding them in, or leaving them to a future update. Our argument for this approach is that if there are only a limited number of relevant additions, it could be reasonable to include them here, but with the field moving so fast we are worried about getting stuck in a situation where adding them in is sufficient to trigger a new round of review, which in turn sets us back long enough for more new additions to be made to the literature, and so on. Since we anticipate a continued high rate of publication in this area, additional review papers in the future to synthesize these new papers will be warranted, with potentially more focused reviews of the most promising research directions.

(lines 8, 9) Please be consistent about the spelling of parameteriz(s)ation

Fixed

In section 1, there are no references at all, even though there are numerous statements that would normally warrant references; this was quite distracting until I understood why. I now understand why this was done, since the references for those statements are given extensively in the sections that follow. My suggestion would be to make a statement early on in the introduction that states something like "In this introduction, we overview the state of machine learning in weather and climate research without providing references; we instead provide references for these statements in the detailed sections that follow."

This is a good suggestion, and we have added words to this effect after the first paragraph of the introduction: "In the remainder of this introduction, we overview the state of machine learning in weather and climate research without always providing references; we instead provide relevant references in the detailed sections that follow."

(line 20) "numerical weather and climate forecasts..are not amenable to transfer to specialized compute resources such as GPUs" … I'm not sure that's strictly true. There's been quite a lot of effort to refactor and port major codes to GPUs, demonstrating that it can be done in principle (for example consider the US Department of Energy's E3SM / SCREAM model, which has a dynamical core that now runs on GPU; https://climatemodeling.science.energy.gov/technical-highlights/simple-cloud-resolving-e3sm-atmosphere-model-scream). It might be more accurate to say that it requires person-decades of effort to transfer these codes to GPUs.

The first referee made a similar observation, and we have amended the text to soften it and acknowledge that it is doable, albeit hard: "These numerical weather and climate forecasts are

computationally costly and are not easy to implement on specialized compute resources such as GPUs (although there are efforts underway to do so, for example in LFRic (Adams et al. 2019)).”

Adams, S. V., Ford, R. W., Hambley, M., Hobson, J. M., Kavčič, I., Maynard, C. M., ... & Wong, R. (2019). LFRic: Meeting the challenges of scalability and performance portability in Weather and Climate models. *Journal of Parallel and Distributed Computing*, *132*, 383-396.

(line 24) “improve the representation of sub grid-scale processes…a computationally costly exercise” <– this isn't necessarily true. Yes, for something like boundary layer turbulence or aerosol physics, modeling higher order moments or doing bin microphysics is more costly. But for something like convection, improvements could come simply through better physics-based theories about how convection works.

The first referee also commented on this. We acknowledge that this is a fair point, and we have modified the text to account for this: “An additional pathway to improve skill is to improve the understanding and representation of sub grid-scale processes, however this is again a potentially computationally costly exercise.”

(lines 106-107) “Furthermore, in many cases…the work is led by data scientists and ML researchers with limited expertise in weather and climate model evaluation” <– this wording risks alienating and insulting colleagues who have done work in this area who in fact have extensive expertise in weather and climate modeling. For example, consider research cited in this paper from the groups of Libby Barnes, Mike Pritchard, and Chris Bretherton – all three of them are definitively experts in weather and climate model evaluation. I suggest revising “in many cases” to “in some cases”.

This is a very good point, and we certainly do not want to undermine the expertise of those contributing to this field who do have expertise in weather and climate model evaluation. We have amended this as you suggest: “Furthermore, in some cases of ML approaches... the work is led by data scientists and ML researchers with limited expertise in weather and climate model evaluation.”

(line 113) “A review of the application of, and progress in, ML in these areas would be of great value…” <– FYI, a review paper by Maria Molina was just accepted in the AMS journal AI4ES, titled “A Review of Recent and Emerging Machine Learning Applications for Climate Variability and Weather Phenomena.” If her paper appears online before this manuscript is finalized, I recommend citing it here. Full disclosure: I'm one of the authors of that paper.

Thank you for drawing this to our attention – it has just appeared in our alerts, and we have included it in the introduction:

“Molina et al. (2023) have provided a very useful review of ML for climate variability and extremes which is highly complementary to this review. They draw similar lines of delineation in the earth system modelling (ESM) value chain to those mentioned above; describing them as “initializing the ESM, running the ESM, and postprocessing ESM output”. They examine each of these steps in turn, with a

focus on the prediction of climate variability and extremes. Here we take a different approach, focusing on one part of the value chain (running the ESM), but looking in more detail at this one part."

It looks like a really useful resource, and it complements this review well.

(line 238) Was GCM defined as an acronym before this? (It's defined later on line 253)

Oops, no it wasn't. Thank you for picking up on this. We have moved the definition to the first instance of "GCM" being used (and removed the definition from further down).

*throughout the paper* There is some odd formatting in the footnotes…they should probably be superscripts. Likewise, the dagger symbol, that indicates a vocab word defined in the glossary, should consistently be a superscript (sometimes it isn't.)

We have attempted to resolve the odd formatting issues. MS Word has been doing some strange things during conversion to PDF but we will take extra care to resolve these issues before our final submission. Hopefully anything we miss will be resolved by the journal's editorial team!

one of my students comments, and I agree: "Personally speaking, I found the paper's pace a bit choppy at times. For example, subsections 3.6-3.9, 5.1, 5.6, and 7.1-7.2 are only a paragraph long. Especially when these occurred back-to-back, the paper felt very "stop go stop go stop go", made even worse when the sequential subsections had little to do with each other. I'm not entirely sure of a solution here, but I wish the authors could find a way to make these subsections flow more together, or at least give us a bit more time with them. It's hard for me to digest their information when each paragraph is immediately moving on to something almost completely different. This could totally just be a me-thing though."

We received similar feedback from the first reviewer, and attempted to address the sense of the sections not flowing from each other with a bit more of an explanation of the logical flow in the introduction. That said, we aren't sure how to address it better than that either – the feeling of choppiness in the short subsections in Section 7 we would argue is due to the topics being touched on having a lot of depth that isn't covered. These subsections are intended to be tasters of these areas which an interested reader could explore further by following the references. To give a better sense of flow between subsections here would probably require expanding them significantly, which we don't feel is feasible for such an already long review. Hopefully the sense of choppiness isn't too off-putting.

Section 7.2: There's a bit more work in this area than just Mudigonda et al. (2017). Here are a few additional relevant papers (again full disclosure: I'm a co-author on two of these):

Prabhat, Kashinath, K., Mudigonda, M., Kim, S., Kapp-Schwoerer, L., Graubner, A., et al. (2021). ClimateNet: an expert-labeled open dataset and deep learning architecture for enabling high-precision analyses of extreme weather. Geoscientific Model Development, 14(1), 107–124. https://doi.org/10.5194/gmd-14-107-2021

O'Brien, T. A., Risser, M. D., Loring, B., Elbashandy, A. A., Krishnan, H., Johnson, J., et al. (2020). Detection of atmospheric rivers with inline uncertainty quantification: TECA-BARD v1.0.1. Geoscientific Model Development, 13(12), 6131–6148. https://doi.org/10.5194/gmd-13-6131-2020

Rupe, A., Kashinath, K., Kumar, N., Crutchfield, J. (2023). Physics-Informed Representation Learning for Emergent Organization in Complex Dynamical Systems. arXiv. https://doi.org/10.48550/arXiv.2304.12586

https://ai4earthscience.github.io/neurips-2020-workshop/papers/ai4earth_neurips_2020_55.pdf

Thank you for pointing these out. It's clear from reading these papers that the area of extreme event identification is one where you have a great deal more expertise than we do!

We have extended the section on object detection (now section 7.4) significantly to include a brief description of each of these papers and have updated the section to take the extra information into account.

We also moved the sub-section further down in section 7, as it seemed to flow more logically that way.